# Epidemiology, mortality, and health service use of local-level multimorbidity patterns in South Spain

Javier Alvarez-Galvez [1,2,3] ✉, Esther Ortega-Martin[1,2], Begoña Ramos-Fiol [1,2], Victor Suarez-Lledo [2,4] & Jesus Carretero-Bravo [1,2]

Multimorbidity –understood as the occurrence of chronic diseases together– represents a major challenge for healthcare systems due to its impact on disability, quality of life, increased use of services and mortality. However, despite the global need to address this health problem, evidence is still needed to advance our understanding of its clinical and social implications. Our study aims to characterise multimorbidity patterns in a dataset of 1,375,068 patients residing in southern Spain. Combining LCA techniques and geographic information, together with service use, mortality, and socioeconomic data, 25 chronicity profiles were identified and subsequently characterised by sex and age. The present study has led us to several findings that take a step forward in this field of knowledge. Specifically, we contribute to the identification of an extensive range of at-risk groups. Moreover, our study reveals that the complexity of multimorbidity patterns escalates at a faster rate and is associated with a poorer prognosis in local areas characterised by lower socioeconomic status. These results emphasize the persistence of social inequalities in multimorbidity, highlighting the need for targeted interventions to mitigate the impact on patients' quality of life, healthcare utilisation, and mortality rates.

We can define multimorbidity as the presence of two or more chronic conditions in an individual[1]. Multimorbidity has become increasingly prevalent worldwide due to the rising incidence of chronic diseases and the ageing population[2,3], which poses significant challenges to healthcare systems[4,5]. This health condition is associated with various negative outcomes, including reduced quality of life, increased disability, functional impairment, increased healthcare utilisation, and fragmented care[6–8]. Treatment of multimorbidity can be complex, leading to poly-medication and higher health and social costs. Individuals with multimorbidity are at a higher risk of experiencing poorer mental and physical health outcomes[9,10] and common mental disorders directly related to multiple physical chronic diseases[3,11]. Addressing multimorbidity is crucial to improving health outcomes and quality of life, reducing healthcare costs, and improving healthcare delivery.

Several studies have pointed out that current clinical guidelines are insufficient in meeting the complex needs of patients with multimorbidity due to inadequate attention to co-occurring diseases[12,13]. As a result, it is necessary to adopt new approaches to address the challenges of multimorbidity. Recently, researchers have focused on identifying specific patterns of chronic conditions that not only consider the accumulation of conditions but also how they cluster in different individuals[14,15]. Several techniques have been employed to detect multimorbidity profiles, among which latent class analysis (LCA) stands out as one of the most commonly utilised[15]. This technique enables the identification of classes of binary variables, allowing

[1]Department of General Economy (Health Sociology area), Faculty of Nursing and Physiotherapy, University of Cadiz, Cadiz, Spain. [2]Computational Social Science DataLab, University Institute for Sustainable Social Development, University of Cádiz, Jerez de la Frontera, Spain. [3]Biomedical Research and Innovation Institute of Cadiz (INiBICA), Hospital Puerta del Mar, Cadiz, Spain. [4]Department of Sociology, University of Granada, Granada, Spain. ✉ e-mail: javier.alvarezgalvez@uca.es

for grouping individuals with similar chronic conditions. By identifying these patterns, clinicians can develop tailored interventions that address the unique needs of patients with multimorbidity.

The multimorbidity patterns commonly described in the literature are as follows: (1) Cardiovascular; (2) Musculoskeletal; (3) Mental; (4) Respiratory; (5) Complex (characterised by many simultaneous chronic diseases linked to different disease patterns); (6) Cancer; (7) Metabolic; and (8) Neurologic (i.e., those with minor or less severe conditions than other people with multimorbidity in their age group)[15,16]. However, although generic patterns are relatively well identified, the wide variability of specific patterns studied in research practice makes it difficult to understand this complex phenomenon as a whole. Thus, there is a need for studies that address the impact of multimorbidity and its determinants in population groups in which this problem remains invisible (e.g., women, children, adolescents and young adults, ethnic groups, and populations with disabilities), as well as additional work with more heterogeneous samples (i.e., that do not focus solely on older persons) and that use robust methodologies for better classification and subsequent understanding of multimorbidity patterns[16]. Indeed, findings are currently limited to samples oriented to specific population groups (generally older population, as this is the group in which chronicity is most prevalent)[17–21], which makes it challenging to obtain a panoramic view of the different patterns and a better measurement of their impact on the health of the population. Therefore, it is crucial to conduct comprehensive studies that aim to identify and characterise diverse multimorbidity patterns along with their associated health outcomes. Furthermore, gaining a deeper understanding of the relationship between these patterns and the socioeconomic and health context to which they are linked is essential.

Similarly, considering that previous studies does not provide a clear description of either the social mechanisms or the ecological effects involved in the association between the local area of residence and multimorbidity patterns, the need for studies that address the social context in which multimorbidity emerges and reproduces itself is evident[22,23]. Despite the growing recognition of multimorbidity as a public health concern, there remains to be more clarity surrounding the social mechanisms and ecological effects that contribute to the emergence and persistence of specific multimorbidity patterns. This knowledge gap highlights the need for studies beyond simply describing the phenomenon and instead focusing on understanding the complex social contexts in which these patterns arise. Furthermore, it is critical to consider how different areas of residence contribute to social inequalities in multimorbidity patterns. By measuring and characterising these inequalities, policymakers could develop targeted interventions and strategies that address the root causes of multimorbidity disparities in health outcomes[24,25].

Using a dataset composed of the health records of 1,375,068 patients treated by the public health system in the province of Cadiz (Spain) until 2021, we aim to characterise the different patterns of multimorbidity and their relationship with service use and mortality. Our specific objectives are: (1) Identify multimorbidity patterns in the subset of pluri-pathological patients using LCA, stratified by sex and age; (2) Analyse geographic variations in the identified patterns, including grouping local health areas by socioeconomic status (SES) and studying the prevalence of multimorbidity classes in these groups; and (3) Measure the impact of different multimorbidity patterns on healthcare utilisation and mortality, considering the SES of the local health area.

## Results

### Prevalence of multimorbidity patterns

Out of the initial dataset of 1,375,068 patients, the final dataset consisted of 1,142,367 individuals residing in the province after excluding non-resident population attended in this area. Among this set of health service users, 490,130 individuals were identified as having

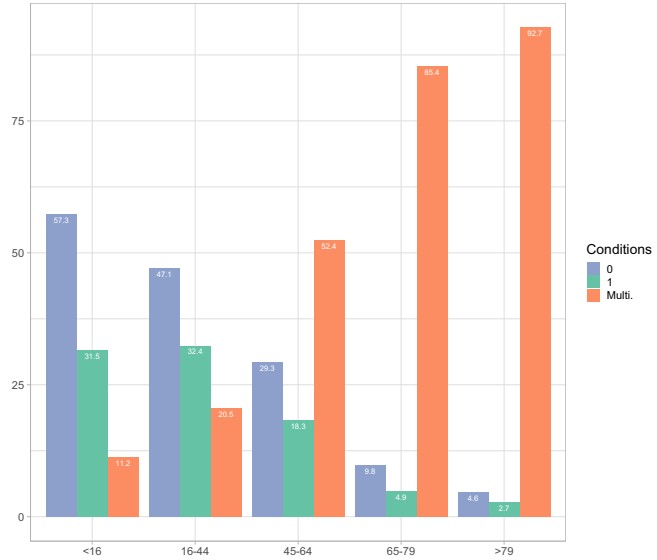

**Fig. 1 | Multimorbidity prevalence by age.** The bars represent the proportion of individuals in each age group without any conditions, with only one condition, and with two or more conditions (multimorbidity). We observe a significant rise in multimorbidity prevalence as age increases.

multimorbidity, resulting in an overall multimorbidity prevalence rate of 42.9%. We observed a range from 11.2% in the <16 age group to 92.7% in the 79+ age group. Furthermore, the mean number of conditions varied across age groups, with the under 16 age group having an average of 0.59 conditions, while the older age group had an average of 5.54 conditions (Fig. 1).

Table 1 presents the main characteristics of the sample by sex group and (average) number of conditions according to the different age groups analysed. The top ten most prevalent conditions are also included.

After identifying individuals with multimorbidity, we employed LCA to derive multimorbidity patterns while considering sex and age groups (<16, 16–44, 45–64, 65–79, >79). To determine the appropriate number of patterns within each age stratum, we relied on three goodness-of-fit indices (BIC, ABIC, and CAIC). The selection analysis aimed to address two key aspects. First, we sought the point at which the goodness-of-fit index ceased decreasing or exhibited a less pronounced decrease. Second, we considered the clinical relevance and interpretability of the resulting patterns. A comprehensive description of the LCA results is provided in the supplementary material (Supplementary Information).

Following a thorough analysis of the goodness-of-fit indices and the resulting combinations of chronic conditions, we were able to identify a total of 25 distinct patterns of multimorbidity. Figure 2 illustrates the relative prevalence of each pattern among individuals with multimorbidity, categorised by age and sex groups. The number of cases per multimorbidity patterns are described in Supplementary Table 1.

Among the obtained patterns, it is worth highlighting the extensive range of combinations observed within this population. Although we identified clearly defined multimorbidity patterns, such as those characterised by diseases linked to the same organ system (e.g., digestive, cardiometabolic, or musculoskeletal), we also observed multimorbidity profiles combining two or three different patterns (e.g., musculoskeletal + cardiovascular, respiratory + cardiovascular, etc.). In addition, we noticed a higher prevalence of patterns involving the accumulation of two or three conditions among middle age groups, whereas patterns became more complex with increasing age. Moreover, we identified four complex multimorbidity patterns, which

**Table 1 | Characteristics of each stratum in the sample of Cadiz Province**

| Characteristic | | <16 | 16–44 | 45–64 | 65–79 | >79 |
|---|---|---|---|---|---|---|
| Gender, n (%) | Men | 88,290 (51.6) | 205,084 (49.52) | 170,840 (48.77) | 69,242 (46.52) | 21,121 (36.45) |
| | Women | 82,817 (48.4) | 209,101 (50.48) | 179,448 (51.23) | 79,606 (53.48) | 36,818 (63.55) |
| Number of morbidities, n (%) | No condition | 98,055 (57.31) | 194,911 (47.06) | 102,576 (29.28) | 14,517 (9.75) | 2659 (4.59) |
| | 1 condition | 53,909 (31.51) | 134,346 (32.44) | 64,247 (18.34) | 7224 (4.85) | 1549 (2.67) |
| | 2 or more | 19,143 (11.19) | 84,928 (20.5) | 183,465 (52.38) | 127,107 (85.39) | 53,731 (92.74) |
| Mean conditions (all people) | | 0.589 | 0.89 | 2.114 | 4.274 | 5.537 |
| Mean conditions (multimorbid) | | 2.448 | 2.761 | 3.686 | 4.948 | 5.942 |
| Top 10 prevalent conditions (%) | | Asthma (15.86) | Asthma (14.65) | Dyslipidaemia (33.35) | Hypertension (64.04) | Hypertension (81.12) |
| | | Atopic dermatitis (8.86) | Anxiety disorder (10.18) | Hypertension (27.89) | Dyslipidaemia (57.82) | Arthrosis, spondylosis (64.26) |
| | | Food intolerance syndrome (4.29) | Dyslipidaemia (7.71) | Arthrosis, spondylosis (21.64) | Arthrosis, spondylosis (51.95) | Dyslipidaemia (56.12) |
| | | Gastro-oesophageal reflux disease (4.24) | Hypothyroidism (6.59) | Anxiety disorder (16.13) | Diabetes (30.9) | Diabetes (38.38) |
| | | Obesity (2.7) | Hypertension (4.41) | Diabetes (10.15) | Anxiety disorder (15.93) | Heart failure (22.95) |
| | | Another developmental disorder (2.28) | Obesity (3.56) | Hypothyroidism (8.99) | Hypothyroidism (12.81) | Osteoporosis (16.79) |
| | | COPD (2.21) | Mood disorder (3.27) | Mood disorder (8) | COPD (12.08) | Atrial fibrillation (16.49) |
| | | Adolescence childhood start disorder (2.14) | Tobacco dependence (3.18) | Asthma (7.83) | Mood disorder (10.1) | COPD (16.44) |
| | | Dyslipidaemia (1.53) | Urinary lithiasis (2.99) | Tobacco dependence (7.3) | Osteoporosis (9.63) | Chronic renal failure (15.04) |
| | | Anxiety disorder (1.52) | Arthrosis, spondylosis (2.86) | Urinary lithiasis (5.69) | Ischaemic heart disease (9.2) | Anxiety disorder (14.56) |

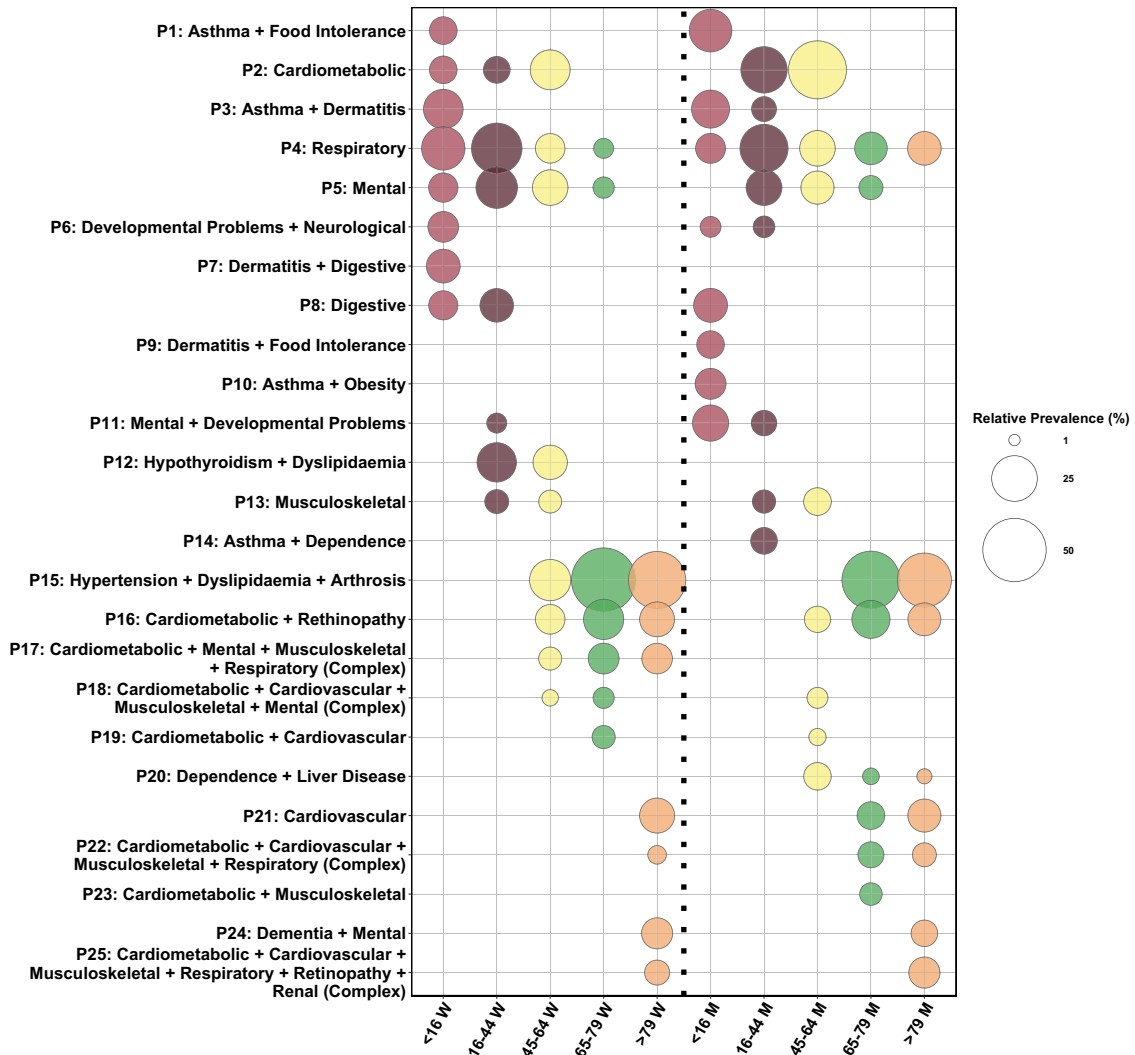

**Fig. 2 | Multimorbidity pattern distribution and prevalence by sex and age.** The left side of the bubble plots show relative prevalence for women, and the right side show the relative prevalence for men.

involve the accumulation of diseases from three or more different patterns or organ systems.

Regarding sex, while the relative prevalence of most patterns is similar within this southern region of Spain, we found some noteworthy differences. Specifically, women had a higher accumulation of patterns associated with mental health, whereas men exhibited patterns related to addictions (Tobacco, Alcohol or Drugs dependence) that were not as prevalent in women. Furthermore, both sexes shared cardiometabolic patterns, yet men had a higher prevalence of such patterns compared to women.

**Multimorbidity patterns by local health area**

Using the geographical information available in the dataset, we obtained the prevalence of patterns according to each local health area in the province of Cadiz. Figures 3 and 4 show the prevalence of the 5 patterns we consider most relevant (P4 [Respiratory]; P5 [Mental]; P15 [Hypertension + Dyslipidaemia + Arthrosis]; P16 [Cardiometabolic + Retinopathy]; P20 [Dependence + Liver Disease]) plus the sum of the prevalence of the 4 localised complex patterns. To better understand each local health area in the province of Cadiz, the maps describe two variables on a dual chromatic scale: the prevalence of the patterns and the income per person (in euros) in each area. Maps of the other 19 patterns can be found in the Supplementary Information (Supplementary Figs. 22–46).

Although in general terms there is no clear spatial relationship in the occurrence of the different patterns of multimorbidity in the province, an inverse relationship between income and multimorbidity can be observed (i.e., the lower the income of the area, the higher the multimorbidity), especially in the greater urban areas such as Algeciras, Cadiz and Jerez. Pattern P4 (Respiratory) exhibits a high prevalence in two urban local health areas of Algeciras, as well as in two semi-rural municipalities. Low-income rural areas show a relative low prevalence of this respiratory pattern. In contrast, Pattern P5 (Mental) demonstrates a medium/low prevalence in city centres, but higher rates are observed in lower-income neighbourhoods of Algeciras and Jerez. Notably, the highest prevalence of this pattern is found in Puerto Serrano, a municipality with the lowest income in both the province and Spain as a whole. On the other hand, Pattern P15 (Hypertension + Dyslipidaemia + Arthrosis) reveals a high prevalence primarily in the city centre of Jerez and in the coastal areas of Cadiz and Algeciras (i.e., high income areas).

P16 (Cardiometabolic + Retinopathy) shows a high prevalence on the south-Atlantic coast of the province and in the centre of some of the main cities. P20 (Dependence + Liver Disease), identified only in men, is more prevalent in low-income neighbourhoods on the periphery of the cities and the northern part of the province (mostly rural areas). Similarly, complex patterns are highly concentrated in low income peripheral and rural areas of the province.

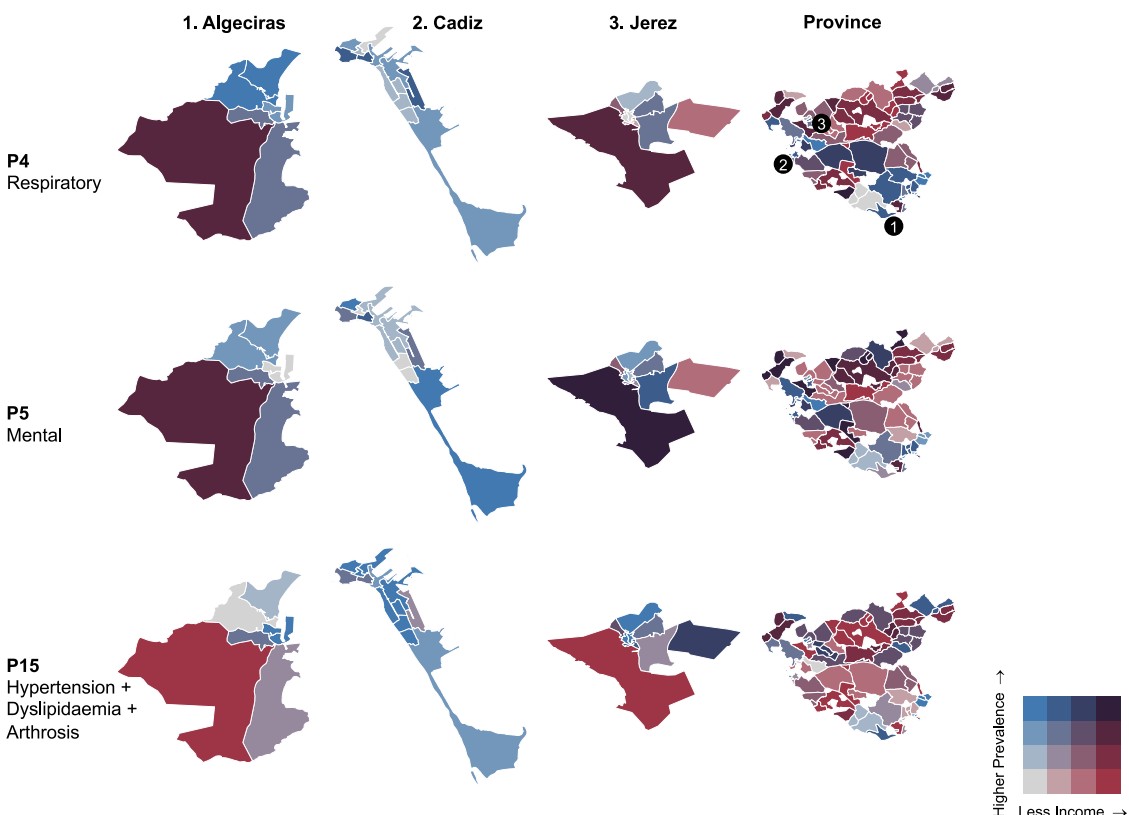

**Fig. 3 | Prevalence of P4 (Respiratory), P5 (Mental) and P15 (Hypertension + Dyslipidaemia + Arthrosis) by local health area.** In the maps, we can see the absolute prevalence of each pattern within all persons with multimorbidity at the level of the whole province (right side map) and the level of the three main urban centres of the province (Cities: 1 Algeciras, 2 Cadiz, and 3 Jerez, in the left side).

Therefore, important differences can be observed in the prevalence of the patterns linked to the area of residence and in the ecological associations with income. In view of these variations at the geographical level, we carried out a cluster analysis to assess the possible similarities between the different geographical areas according to their SES. Thus, we used two ecological variables, income per person and the deprivation index, to classify the different areas, which were finally grouped into three clusters: low SES, medium SES and high SES. The prevalence of patterns according to each age group would then be compared in relation to these SES areas. Table 2 shows the differences in the prevalence of patterns in women, and Table 3 shows the differences in the group of men. We can observe in Table 2 how health areas with low SES have a higher prevalence of patterns P5 (Mental), P16 (Cardiometabolic + Retinopathy), P17 (Complex), P18 (Complex) and P21 (Cardiovascular) in women. On the other hand, areas with high SES have a higher prevalence of patterns P4 and P15, patterns with fewer and highly prevalent diseases such as hypertension, dyslipidaemia, and respiratory conditions.

In Table 3, which shows the differences in men, patterns P2 (Cardiometabolic), P9 (Dermatitis + Food Intolerance), P13 (Musculoskeletal), and P20 (Dependence + Liver Disease) have a higher prevalence in low SES health areas. P5 (Mental) and P15 (Hypertension + Dyslipidaemia + Arthrosis) have a higher prevalence in areas with higher SES, while the P4 (Respiratory) pattern has a higher occurrence of high SES at younger ages and, particularly, in the older low SES group.

In general, we can observe that low SES women are affected by a higher variety of multimorbidity patterns. In addition, while P5 (Mental) have been found to be more prevalent women low SES women, while in men is more prevalent in those living in high SES areas.

## Public health services use and mortality

Finally, we also analysed the relationship between multimorbidity patterns and the average service use, measuring service use with the mean number of visits to general practitioners (GPs) per year, the mean number of visits to hospital visits (i.e., consults with medical specialists) per year and the mean number of visits to A&E units per year (hospitalisations were not included). Figure 5 shows the distribution of service use in people with multimorbidity by sex, age and SES area.

Figure 5 shows that women and low SES groups have a higher use of health services (except for hospital consultations), a difference that is even more pronounced for women. It is also observed that older multimorbidity groups are more likely to use primary care services, while younger multimorbidity groups are more likely to use emergency services. We must consider that the group of children under 16 years of age includes routine paediatric visits, which might explain the higher number of average visits to GPs compared with other groups.

Focusing on specific multimorbidity patterns, Fig. 6 describes the use of health services in each multimorbidity pattern by the SES area. As can be observed, the average number of visits to GPs is similar across SES areas in patterns with fewer conditions, however, as patterns become more complex (P25, P22 or P18, i.e., multi-system patterns) individuals from the lower SES have a significantly higher number of visits to GPs. This is also evident for the A&E visits: the greater the complexity of the patterns, the greater the use of A&E services in the lower SES strata. Nevertheless, we also observe that patterns with younger people and fewer conditions, such as P7, P9 or P11 (characterised by diseases like dermatitis or digestive problems), have a high use of these health services. The only pattern with a high use of A&E services among mid/high SES groups is P11, a pattern of people under 45 years of age with problems mainly associated with

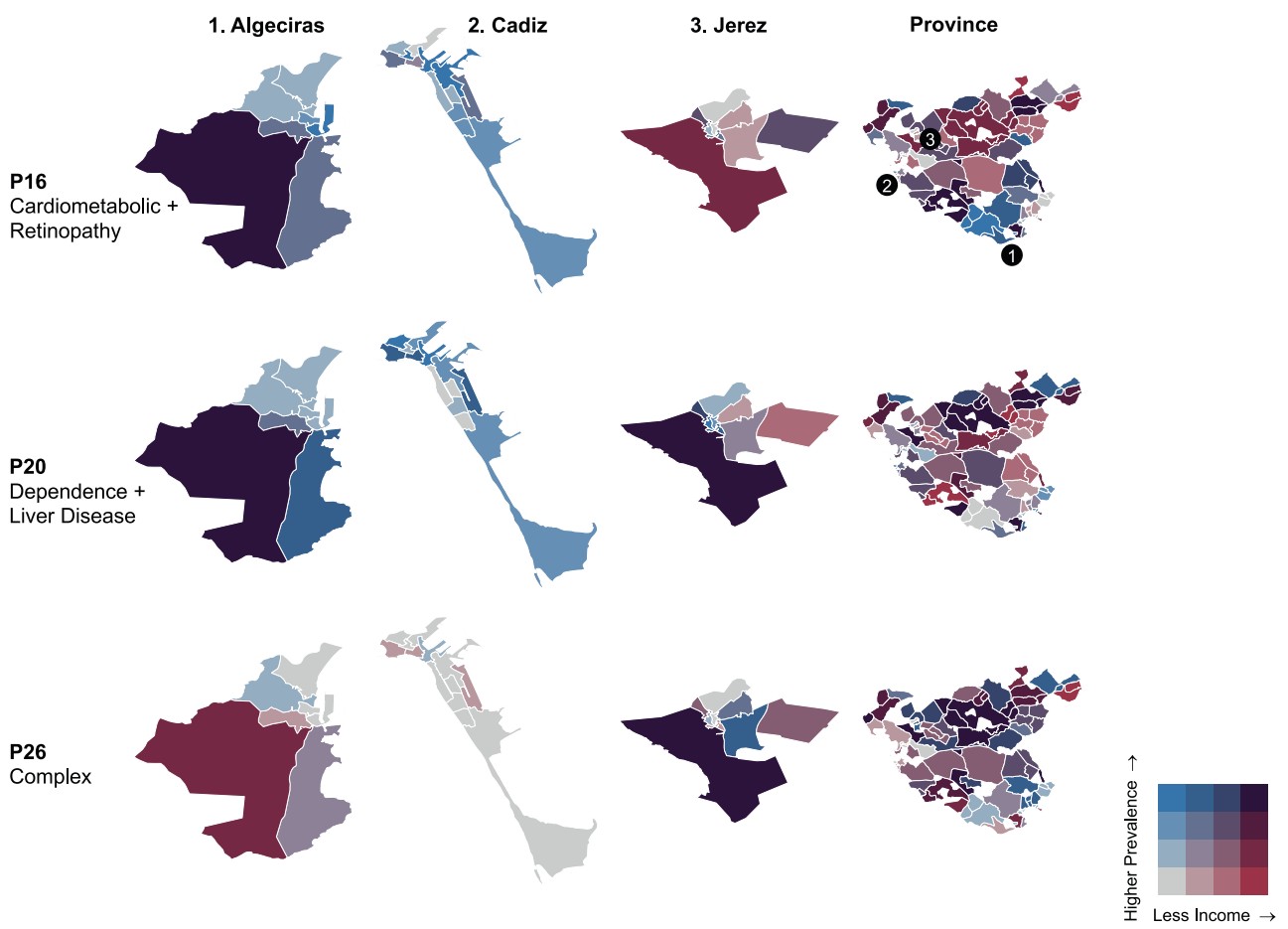

**Fig. 4 | Prevalence of P16 (Cardiometabolic + Retinopathy), P20 (Dependence + Liver Disease) and complex patterns by local health area.** In the maps, we can see the absolute prevalence of each pattern within all persons with multimorbidity at the level of the whole province (right side map) and the level of the three main urban centres of the province (Cities: 1 Algeciras, 2 Cadiz, and 3 Jerez, in the left side).

mental development. Finally, in the case of hospital visits, we can detect the opposite trend. Local health areas with low SES make significantly less use of this type of health service in all patterns, except in pattern P6, formed by younger people with developmental and neurological problems.

To conclude our analysis, we studied the relationship between different patterns of multimorbidity and mortality using logistic regressions in the three SES areas. We discarded patterns P1 (Asthma + Food Intolerance), P3 (Asthma + Dermatitis), P6 (Developmental Problems + Neurological), P7 (Dermatitis + Digestive), P8 (Digestive), P9 (: Dermatitis + Food Intolerance), P10 (Asthma + Obesity), P11 (Mental + Developmental Problems) and P14 (Asthma + Dependence) for this analysis because they had less than 15 deaths by the end of 2021. Figure 7 shows the results of the models in Odds Ratios (95% CI), using P2 (Cardiometabolic) as the reference pattern since it is a transversal class in all strata (Supplementary Information, Supplementary Table 2).

In general, we can observe that mortality is significantly higher in the complex multimorbidity patterns with older people (P18, P25, P22 or P24). However, when examining the disparities by SES areas, the P20 pattern, linked to addictions (alcohol and drug dependence), is found to be the pattern with the highest mortality, despite the fact that is made up of younger men. On the contrary, in mid/high SES areas, P15 (Hypertension + Dyslipidaemia + Arthrosis) is negatively related with mortality compared to the reference pattern (P2: Cardiometabolic).

Additionally, P19, predominantly comprising cardiovascular diseases, demonstrates a stronger effect in mortality in low SES areas.

## Discussion

The detailed characterisation of multimorbidity patterns according to local health areas has allowed us both to provide a comprehensive description of the epidemiology of this health condition and to measure the association of different multi-pathology profiles on health service utilisation and mortality in the province of Cadiz, Spain.

The present study has led us to several findings that take a step forward in this field of knowledge. Using a large dataset with significant heterogeneity of chronic conditions (including addictions) has allowed us to detect 25 multimorbidity patterns, an important advance in the identification of chronicity profiles that are less frequent in the existing literature. Other studies in our country have found fewer multimorbidity profiles because they are based either on fewer chronic conditions or because they have chosen a sample mainly focusing on older individuals[18,26–31]. On the other hand, our study highlights the escalating complexity of multimorbidity as the population ages. Nevertheless, the prevalence of multimorbidity exceeding 20% within the 16–44 age group highlights the necessity for comprehensive studies that cover all age groups, rather than solely focusing on older population groups.

At the methodological level, it is worth underlining the fact that we have performed a differentiated LCA analysis in each sex and age

**Table 2 | Differences in the prevalence of multimorbidity patterns by SES area (women)**

| Characteristic | Age group | Low SES, N = 12[a] | Medium SES, N = 34[a] | High SES, N = 22[a] | p value[b] |
|---|---|---|---|---|---|
| P1: Asthma + Food Intolerance | <16 | 6.20 (3.82, 7.78) | 7.23 (4.69, 9.51) | 7.35 (5.26, 9.22) | 0.45 |
| P2: Cardiometabolic | <16 | 10.62 (7.16, 14.45) | 9.12 (5.20, 11.28) | 7.35 (5.89, 9.41) | 0.12 |
| | 16–44 | 8.47 (6.79, 9.37) | 8.09 (7.12, 8.97) | 7.54 (6.42, 9.55) | 0.95 |
| | 45–64 | 17.94 (15.93, 19.32) | 18.93 (18.06, 20.26) | 17.86 (17.13, 19.86) | 0.091^ |
| P3: Asthma + Dermatitis | <16 | 15.96 (10.64, 20.96) | 18.65 (13.27, 23.74) | 17.66 (14.07, 23.20) | 0.79 |
| P4: Respiratory | <16 | 17.57 (13.64, 23.12) | 22.59 (19.44, 27.82) | 21.87 (20.25, 25.87) | 0.19 |
| | 16–44 | 23.98 (21.94, 25.76) | 30.37 (29.04, 33.08) | 31.70 (29.64, 33.86) | <0.001*** |
| | 45–64 | 6.36 (5.72, 7.97) | 9.54 (8.39, 10.79) | 10.50 (9.79, 11.14) | <0.001*** |
| | 65–79 | 3.61 (3.02, 4.46) | 3.92 (3.38, 4.45) | 3.84 (3.44, 5.44) | 0.57 |
| P5: Mental | <16 | 8.30 (7.24, 11.16) | 9.54 (6.95, 11.52) | 7.88 (6.93, 10.72) | 0.57 |
| | 16–44 | 22.88 (21.27, 26.51) | 19.77 (17.51, 22.18) | 20.23 (17.25, 21.02) | 0.011* |
| | 45–64 | 14.80 (12.60, 15.89) | 13.57 (12.38, 14.50) | 15.13 (13.37, 15.73) | 0.047* |
| | 65–79 | 5.28 (4.32, 5.65) | 4.60 (4.06, 5.13) | 4.44 (3.59, 4.91) | 0.25 |
| P6: Developmental Problems + Neurological | <16 | 11.43 (7.14, 18.06) | 10.29 (7.36, 13.02) | 9.96 (7.66, 12.82) | 0.7 |
| P7: Dermatitis + Digestive | <16 | 11.94 (2.06, 21.41) | 11.57 (8.74, 14.62) | 13.05 (10.28, 16.26) | 0.46 |
| P8: Digestive | <16 | 8.95 (6.74, 13.37) | 9.62 (5.24, 13.37) | 8.43 (4.91, 13.37) | 0.88 |
| | 16–44 | 15.25 (11.74, 16.42) | 12.57 (11.56, 14.48) | 13.16 (12.14, 14.46) | 0.26 |
| P11: Mental + Developmental Problems | 16–44 | 3.77 (2.60, 5.05) | 3.53 (2.95, 4.34) | 4.28 (3.72, 5.26) | 0.065^ |
| P12: Hypothyroidism + Dyslipidaemia | 16–44 | 18.42 (15.00, 19.65) | 18.96 (14.92, 22.21) | 16.73 (14.30, 21.27) | 0.81 |
| | 45–64 | 13.30 (12.14, 14.88) | 12.89 (11.82, 14.12) | 14.24 (13.34, 15.66) | 0.044* |
| P13: Musculoskeletal | 16–44 | 5.94 (5.60, 8.32) | 6.46 (5.41, 7.52) | 5.70 (4.72, 6.70) | 0.24 |
| | 45–64 | 5.44 (4.26, 6.99) | 5.74 (5.18, 6.73) | 5.36 (4.95, 5.64) | 0.17 |
| P15: Hypertension + Dyslipidaemia + Arthrosis | 45–64 | 20.91 (19.36, 23.04) | 20.21 (19.26, 22.07) | 19.24 (18.66, 20.57) | 0.065 |
| | 65–79 | 46.19 (42.73, 48.26) | 49.84 (48.01, 52.58) | 54.62 (51.84, 57.98) | <0.001*** |
| | >79 | 37.77 (32.60, 40.26) | 39.38 (35.09, 42.58) | 45.09 (38.61, 47.91) | 0.001** |
| P16: Cardiometabolic + Retinopathy | 45–64 | 10.55 (9.46, 11.50) | 9.88 (9.23, 11.55) | 9.47 (8.44, 10.22) | 0.07^ |
| | 65–79 | 19.97 (18.59, 22.28) | 20.45 (18.89, 21.33) | 17.30 (15.18, 19.09) | <0.001*** |
| | >79 | 14.67 (14.40, 15.50) | 14.51 (13.29, 15.73) | 12.57 (11.45, 14.04) | 0.01** |
| P17: Cardiometabolic + Mental + Musculoskeletal + Respiratory (Complex) | 45–64 | 7.14 (5.83, 7.92) | 5.23 (4.34, 6.12) | 5.28 (4.44, 5.84) | 0.009** |
| | 65–79 | 13.04 (11.87, 13.91) | 10.45 (8.62, 11.53) | 9.73 (8.41, 11.20) | 0.001** |
| | >79 | 11.64 (9.47, 13.67) | 10.07 (8.29, 12.00) | 9.51 (8.01, 11.96) | 0.18 |
| P18: Cardiometabolic + Cardiovascular + Musculoskeletal + Mental (Complex) | 45–64 | 2.01 (1.88, 2.46) | 2.67 (1.93, 3.37) | 2.62 (2.03, 3.15) | 0.25 |
| | 65–79 | 5.44 (4.48, 6.42) | 5.02 (4.18, 5.67) | 3.95 (3.46, 4.16) | <0.001*** |
| P19: Cardiometabolic + Cardiovascular | 65–79 | 6.04 (5.10, 6.93) | 5.91 (5.03, 6.36) | 5.40 (5.02, 5.84) | 0.08^ |
| P21: Cardiovascular | >79 | 14.52 (13.76, 15.92) | 14.32 (13.32, 15.50) | 13.23 (12.43, 14.05) | 0.038* |
| P22: Cardiometabolic + Cardiovascular + Musculoskeletal + Respiratory (Complex) | >79 | 3.62 (2.62, 4.93) | 3.26 (2.36, 4.25) | 3.31 (2.61, 3.80) | 0.58 |
| P24: Dementia + Mental | >79 | 10.36 (8.48, 11.38) | 10.89 (9.39, 12.09) | 9.41 (8.54, 12.22) | 0.53 |
| P25: Cardiometabolic + Cardiovascular + Musculoskeletal + Respiratory + Retinopathy + Renal (Complex) | >79 | 6.51 (5.42, 9.01) | 6.89 (5.52, 7.98) | 6.70 (4.87, 8.02) | 0.8 |

*, **, ***, and ^ indicate significance level at 5%, 1%, 0.1%, and 10%, respectively.
[a]Median (IQR).
[b]Kruskal–Wallis rank sum test.

stratum, unlike other studies using the same statistical technique, which is advisable when the difference between the groups in the indicator variables (chronic conditions) can cause the latent variables (multimorbidity profiles) to be structurally different[32]. This has allowed us to detect specific patterns in different age and sex group, and also focus on chronic conditions that have a higher prevalence, which contributes to the creation of more parsimonious multimorbidity classes by discarding indicator variables unrelated to the latent structure.

While our study successfully identified the most prevalent patterns of multimorbidity, including cardiovascular, cardiometabolic, musculoskeletal, mental, and complex patterns in older individuals, it is important to note the detection of additional mixed patterns that

may serve as prodromal indicators. These combined patterns have the potential to evolve into highly complex patterns at older ages. For instance, pattern P2, characterised as cardiometabolic in individuals under 65, may progress into cardiovascular and neurological patterns. Similarly, pattern P20 (Dependence + Liver Disease), observed in men with dependence and liver disease, has the potential to lead to renal issues, cardiovascular problems, cancer, or dementia. These findings emphasise the significance of early detection and intervention strategies to mitigate the development of intricate multimorbidity patterns.

The comprehensive analysis conducted in our study enables us to observe the age-related evolution of patterns. Among younger groups, mental disorders, asthma (including respiratory patterns), dermatitis, and digestive disorders emerge as the primary chronic conditions that

**Table 3 | Differences in the prevalence of multimorbidity patterns by SES area (men)**

| Characteristic | Age group | Low SES, N = 12[a] | Medium SES, N = 34[a] | High SES, N = 22[a] | p value[b] |
|---|---|---|---|---|---|
| P1: Asthma + Food Intolerance | <16 | 21.72 (15.92, 32.33) | 21.21 (16.03, 26.03) | 20.41 (17.88, 25.90) | 0.89 |
| P2: Cardiometabolic | 16–44 | 30.37 (25.71, 32.83) | 26.59 (24.25, 28.30) | 24.78 (23.54, 27.00) | 0.055^ |
|  | 45–64 | 44.75 (42.76, 49.52) | 42.41 (38.71, 44.88) | 40.63 (39.22, 43.01) | 0.014* |
| P3: Asthma + Dermatitis | <16 | 12.25 (9.41, 17.10) | 14.89 (9.77, 21.85) | 16.30 (12.42, 19.52) | 0.32 |
|  | 16–44 | 5.08 (4.17, 6.13) | 5.68 (3.94, 7.06) | 6.72 (5.54, 8.55) | 0.051^ |
| P4: Respiratory | <16 | 6.82 (4.06, 12.23) | 9.82 (5.77, 17.50) | 11.10 (7.99, 13.88) | 0.15 |
|  | 16–44 | 22.84 (18.27, 27.37) | 27.69 (26.02, 30.25) | 29.11 (25.90, 30.79) | 0.031* |
|  | 45–64 | 12.96 (12.08, 13.53) | 14.20 (13.09, 15.40) | 14.99 (14.17, 16.04) | 0.012* |
|  | 65–79 | 11.91 (10.95, 12.96) | 11.88 (10.29, 13.78) | 11.52 (10.33, 14.10) | 0.85 |
|  | >79 | 15.54 (12.66, 18.58) | 13.05 (11.75, 14.86) | 11.19 (9.54, 13.51) | 0.007** |
| P5: Mental | 16–44 | 14.54 (12.74, 17.60) | 14.29 (13.41, 16.34) | 14.47 (13.29, 15.24) | 0.81 |
|  | 45–64 | 9.68 (8.86, 11.16) | 11.55 (10.59, 12.89) | 13.30 (12.19, 14.30) | <0.001*** |
|  | 65–79 | 5.98 (5.54, 6.66) | 5.93 (5.28, 6.66) | 6.07 (5.51, 7.22) | 0.58 |
| P6: Developmental Problems + Neurological | <16 | 4.03 (2.31, 7.12) | 5.06 (3.70, 7.02) | 3.94 (3.08, 5.46) | 0.3 |
|  | 16–44 | 4.14 (3.77, 5.16) | 4.83 (4.19, 5.42) | 4.49 (3.82, 4.86) | 0.26 |
| P8: Digestive | <16 | 11.98 (8.00, 23.89) | 12.09 (5.79, 19.18) | 11.98 (6.56, 16.93) | 0.61 |
| P9: Dermatitis + Food Intolerance | <16 | 10.75 (8.48, 12.71) | 7.65 (5.97, 9.14) | 8.41 (6.17, 11.02) | 0.044* |
| P10: Asthma + Obesity | <16 | 6.32 (4.51, 11.66) | 8.26 (6.19, 13.62) | 9.48 (5.36, 10.86) | 0.59 |
| P11: Mental + Developmental Problems | <16 | 13.22 (9.63, 15.41) | 15.19 (11.51, 18.18) | 16.31 (11.62, 19.35) | 0.51 |
|  | 16–44 | 7.61 (5.98, 8.67) | 6.38 (5.47, 7.63) | 6.84 (6.01, 8.10) | 0.43 |
| P13: Musculoskeletal | 16–44 | 5.64 (5.03, 9.20) | 5.76 (4.99, 7.07) | 5.51 (4.40, 6.04) | 0.2 |
|  | 45–64 | 8.98 (8.13, 11.26) | 8.53 (7.78, 9.50) | 7.83 (7.16, 8.23) | 0.008** |
| P14: Asthma + Dependence | 16–44 | 7.69 (6.43, 9.39) | 7.16 (6.00, 8.38) | 7.38 (6.34, 9.67) | 0.58 |
| P15: Hypertension + Dyslipidaemia + Arthrosis | 65–79 | 41.38 (37.62, 44.55) | 40.52 (37.41, 43.11) | 41.74 (38.84, 44.80) | 0.41 |
|  | >79 | 32.35 (25.89, 35.43) | 34.83 (32.27, 38.34) | 36.66 (34.20, 39.28) | 0.034* |
| P16: Cardiometabolic + Retinopathy | 45–64 | 6.58 (5.63, 7.12) | 7.48 (6.87, 8.40) | 7.27 (6.50, 7.83) | 0.056^ |
|  | 65–79 | 16.35 (15.78, 17.23) | 17.15 (15.87, 18.10) | 16.73 (14.48, 17.94) | 0.43 |
|  | >79 | 12.70 (10.79, 15.12) | 11.30 (10.00, 12.90) | 12.73 (11.01, 14.18) | 0.11 |
| P18: Cardiometabolic + Cardiovascular + Musculoskeletal + Mental (Complex) | 45–64 | 4.52 (3.88, 5.22) | 4.36 (3.92, 5.18) | 4.33 (3.92, 5.26) | 0.97 |
| P19: Cardiometabolic + Cardiovascular | 45–64 | 2.88 (2.29, 3.57) | 2.69 (2.43, 3.16) | 2.91 (2.70, 3.16) | 0.59 |
| P20: Dependence + Liver Disease | 45–64 | 7.87 (6.51, 8.36) | 8.01 (6.21, 10.26) | 8.13 (6.92, 9.30) | 0.79 |
|  | 65–79 | 3.04 (2.16, 3.80) | 2.72 (1.86, 3.42) | 2.22 (1.68, 2.44) | 0.024* |
|  | >79 | 2.60 (2.02, 3.04) | 1.68 (1.12, 2.55) | 1.75 (1.37, 2.55) | 0.2 |
| P21: Cardiovascular | 65–79 | 8.17 (7.32, 9.11) | 8.54 (7.50, 9.12) | 8.64 (8.48, 9.47) | 0.45 |
|  | >79 | 11.56 (9.91, 12.60) | 12.81 (10.83, 14.55) | 12.15 (11.12, 13.78) | 0.22 |
| P22: Cardiometabolic + Cardiovascular + Musculoskeletal + Respiratory (Complex) | 65–79 | 6.72 (5.60, 8.02) | 7.93 (6.79, 8.65) | 7.10 (6.50, 8.36) | 0.49 |
|  | >79 | 7.02 (5.95, 8.09) | 5.46 (4.65, 6.79) | 6.41 (5.30, 7.52) | 0.18 |
| P23: Cardiometabolic + Musculoskeletal | 65–79 | 5.70 (4.90, 6.56) | 5.47 (4.64, 6.43) | 5.10 (4.40, 5.48) | 0.11 |
| P24: Dementia + Mental | >79 | 8.66 (6.10, 9.60) | 7.79 (6.12, 8.84) | 7.62 (6.57, 8.44) | 0.59 |
| P25: Cardiometabolic + Cardiovascular + Musculoskeletal + Respiratory + Retinopathy + Renal (Complex) | >79 | 10.80 (8.10, 14.38) | 12.23 (9.54, 12.96) | 11.62 (8.31, 13.08) | 0.74 |

*, **, ***, and ^ indicate significance level at 5%, 1%, 0.1%, and 10%, respectively.
[a]Median (IQR).
[b]Kruskal–Wallis rank sum test.

contribute to the formation of subsequent multimorbidity profiles. The emergence of the cardiometabolic pattern is evident even in the youngest age groups, underscoring the challenges posed by childhood obesity[33,34], as well as the musculoskeletal pattern. In terms of gender differences, women exhibit a higher prevalence of mental disorders[35,36], whereas men stand out for their propensity towards the cardiometabolic pattern and addiction (particularly related to alcohol and tobacco), which manifest prior to the age of 44 years old[37–40].

As age advances, cardiovascular, musculoskeletal, and respiratory conditions gain prominence, appearing in complex patterns that mix more than three organic systems. It is important to add that some of the patterns found are formed by specific diseases of ageing, without deriving in other more serious chronic conditions (P15: Hypertension + Dyslipidaemia + Arthrosis). Although this pattern is formed by individuals whose age only presents highly prevalent chronic diseases and, in principle, might appear to be a less aggressive pattern for individual health, we must consider that the conjunction of arterial hypertension and dyslipidaemia are the two risk factors with the greatest weight in the development of cardiovascular disease. Thus, this is a pattern that usually appears in other studies of multimorbidity in the population

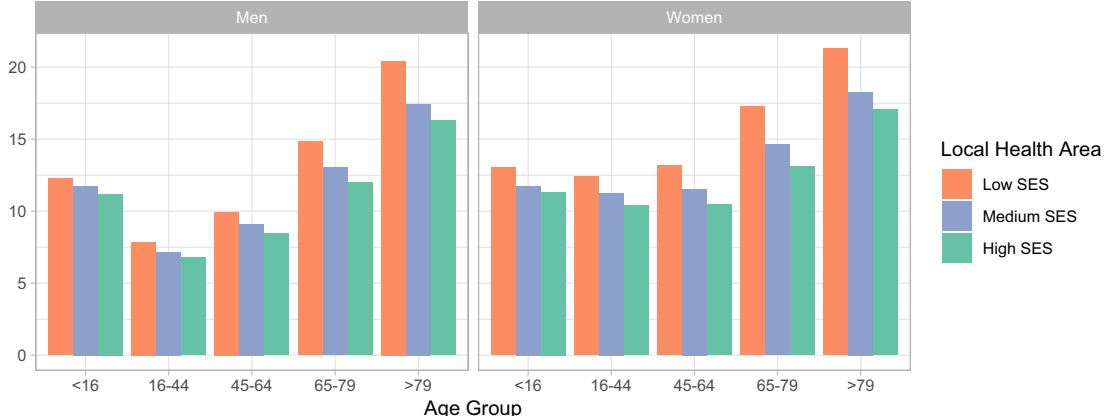

**a**   Mean number of GP visits by sex, age, and SES area

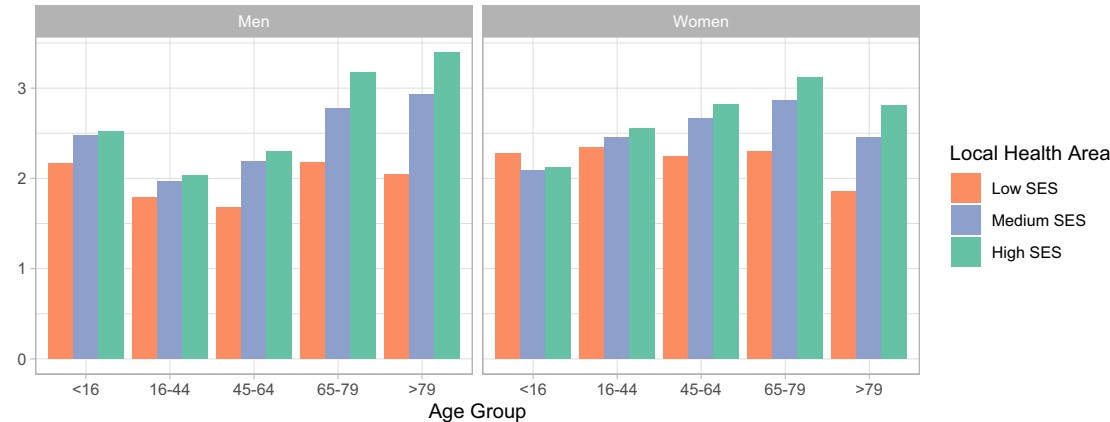

**b**   Mean number of hospital visits by sex, age, and SES area

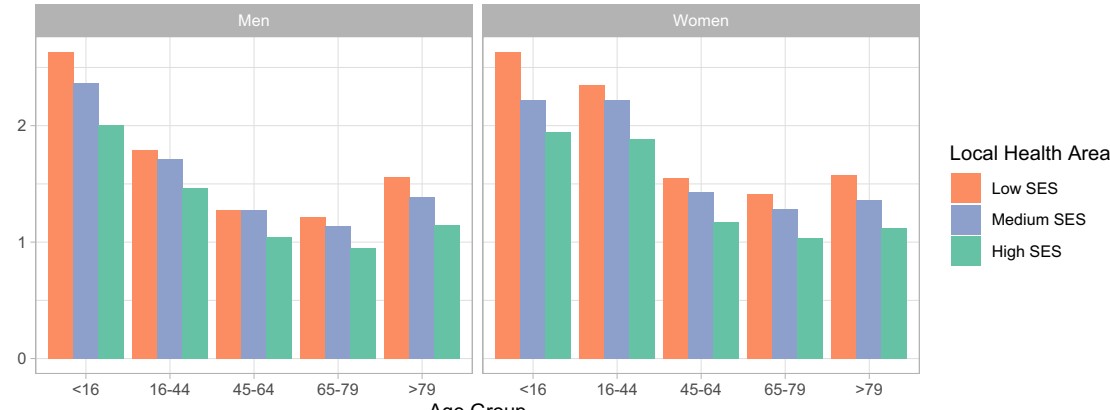

**c**   Mean number of A&E visits by sex, age, and SES area

**Fig. 5 | Mean number of visits to health services of patients with multi-morbidity by sex, age and SES area. a** Mean number of general practitioner (GP) visits by sex, age, and socioeconomic status (SES) area; **b** mean number of hospital visits by sex, age, and SES area; **c** mean number of accidents and emergencies (A&E) visits by sex, age, and SES area.

over 60 years old and which is linked to the growing increase in sedentary lifestyles and obesity[41–45].

At the geographical level, there were large differences in the prevalence of the multimorbidity patterns identified. For example, musculoskeletal patterns are more prevalent in the rural areas of the province, whose labour market historically is characterised by manual occupations in the primary sector in the province (associated with wine industry and agriculture)[46–48], while in urban areas there is a higher prevalence of respiratory multimorbidity profiles, which are generally more common among the young population. These urban

areas of the province are characterised by higher pollution[49,50], especially in the Bay of Algeciras.

Furthermore, depending on the SES of the area of residence, essential differences in multimorbidity patterns were also observed, with mental health and dependence profiles emerging more regularly in the most deprived areas, particularly in older men. This trend is observed in rural areas with lower SES (such as Puerto Serrano, the area with the lowest GDP per capita) and in the poorest neighbourhoods of cities such as Algeciras, Cadiz, or Jerez. In fact, low SES peripheral urban areas are more similar in the prevalence of patterns to rural and

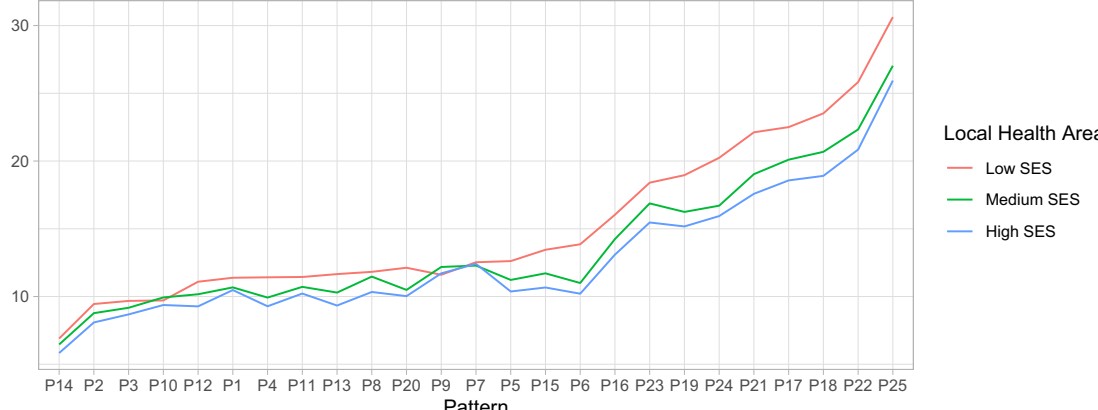

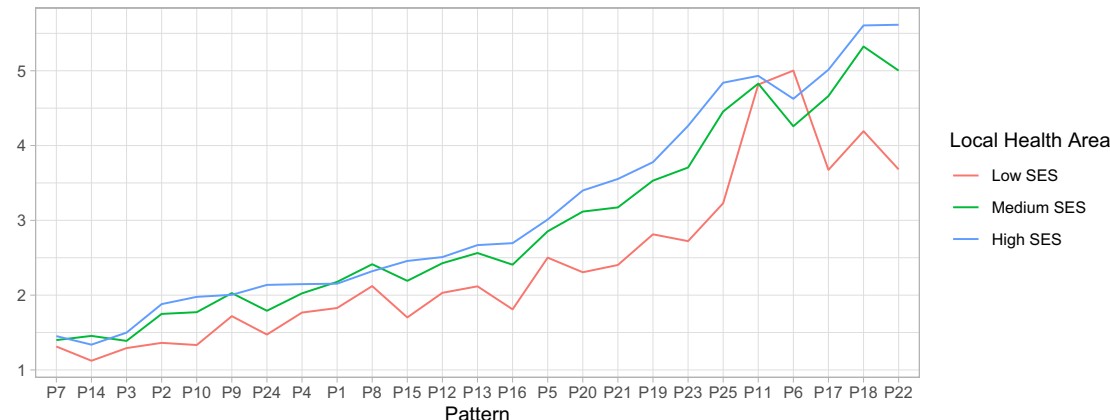

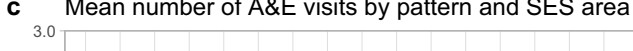

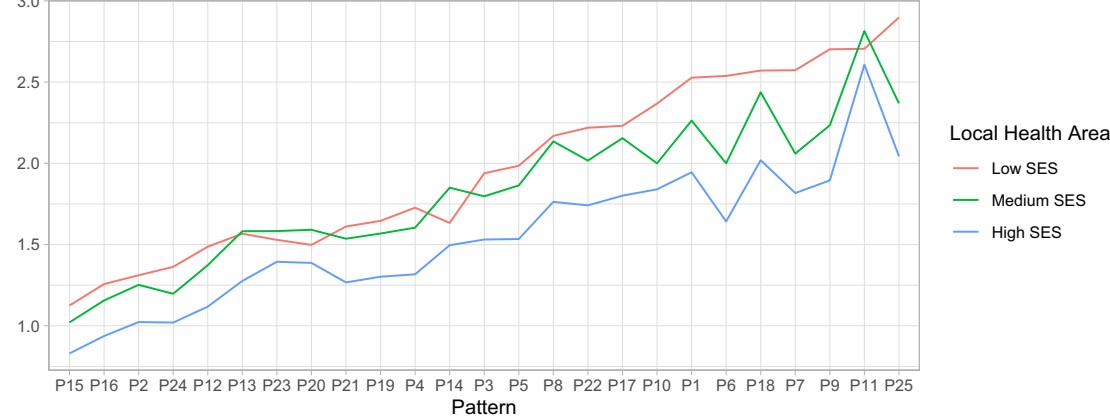

**Fig. 6 | Health services uses by multimorbidity patterns and SES status of local health area. a** Mean number of general practitioner (GP) visits by multimorbidity pattern and socioeconomic status (SES) area; **b** mean number of hospital visits by multimorbidity pattern and SES area; **c** mean number of accidents and emergencies (A&E) visits by multimorbidity pattern and SES area.

impoverished areas of the province than to the urban centres of the cities. It is also noted that, in these areas, mental health patterns (P5) may also have a direct relationship with mortality, even approaching the level of certain cardiovascular patterns (once disaggregated into specific population groups). It is therefore essential to consider possible gaps in mental health care for patients with mental multimorbidity, especially in the most deprived areas, where these problems could have a greater impact on the health of the population.

Among the 45–64 age groups residing in low SES areas, the transformation of the P15 pattern (Hypertension + Dyslipidaemia + Arthrosis) into more severe cardiovascular and complex

multimorbidity profiles is noteworthy, while this same pattern maintains a certain stability in high SES areas in the population beyond 65 years of age. In fact, it is generally observed that the severity and complexity of the patterns is higher in low SES areas. In other words, while the ageing pattern evolves faster and has a worse prognosis in local areas of lower SES, it remains relatively constant in high SES areas. The findings from other studies appear to support this trend of a more adverse prognosis for multimorbidity among individuals with low SES[51,52]. As a result, it is plausible that low SES areas are not only linked to more complex and harmful multimorbidity patterns in later life but also with earlier onset of these same disease combinations.

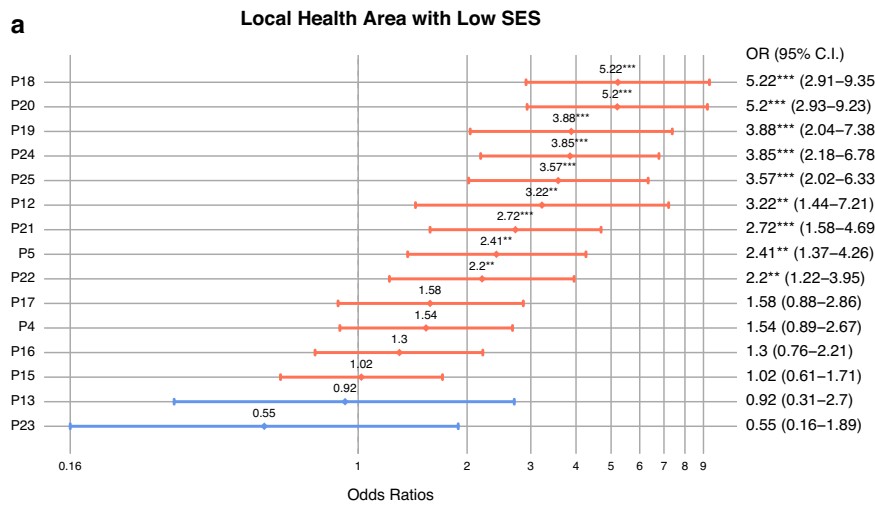

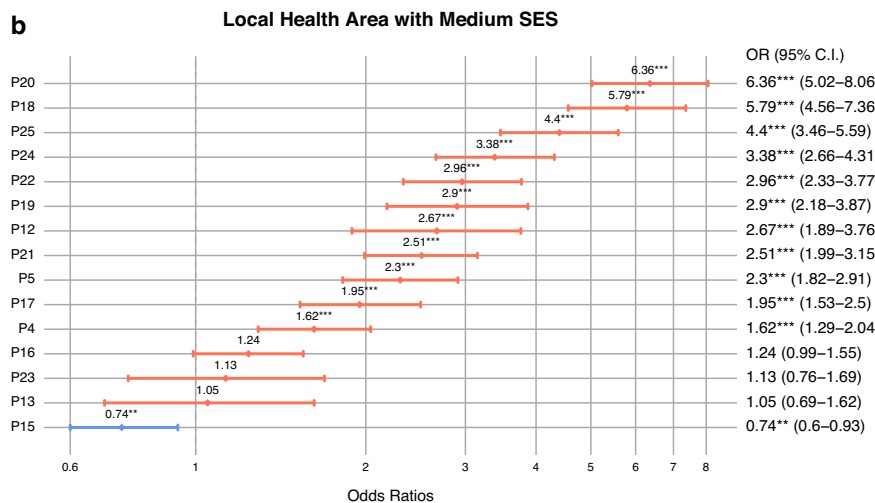

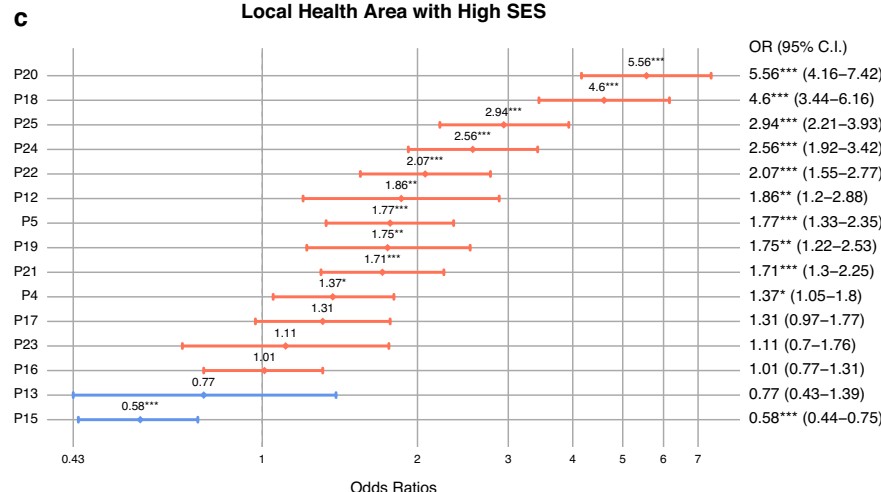

**Fig. 7 | Impact on mortality of the multimorbidity patterns by SES area.** Logistic regression models adjusted by sex and age. **a** Association of multimorbidity pattern with mortality in low socioeconomic status (SES) areas (*N* = 49,939); **b** association of multimorbidity pattern with mortality in medium SES areas (*N* = 256,133);

**c** association of multimorbidity pattern with mortality in high SES areas (*N* = 187,058). Data are presented as odd ratio (OR) with a confidence interval (CI) of 95%.

As expected, we also observed differences in the use of services and mortality. On the one hand, a higher use of health services could be observed among women, older and younger age groups. The fact that women and some younger age groups have a higher use of certain health services may be partly explained by the fact that routine paediatric visits for children under 16 and routine check-ups associated with pregnancy and gynaecology for women are included. However, it should be noted that the higher use by these multi-pathological groups may also reflect that they are not receiving the correct management and/or treatment for their conditions throughout the health care pathway. This is a fundamental point that will need to be addressed in depth in future studies.

It was observed a higher use of health services as the patterns become more complex. The higher prevalence of complex patterns in low SES areas was also associated with a higher frequentation of health services (mainly primary care and, to a lesser extent, A&E), which may be due both to the greater use of private services (including alternative therapies outside the health system) by people of higher SES and to their higher (digital) health literacy compared to people of lower status[53–55]. In any case, although we know that there may be certain differences in the use of health services in relation to the aforementioned aspects, it is necessary to indicate that taking into account, on the one hand, the universality of the Andalusian health system and, on the other, the existence of a single clinical history of the patient in this region, we assume that these variations have more to do with social inequalities in health than with the specific use of public or private health services[56].

These differences were inverse in the number of hospital consults, being lower in all patterns in the low SES areas, except for pattern P6, associated with young people with developmental problems. This could be explained by the difficulties for the population in rural areas to access specialised hospital care, as these geographical areas are further away from the main hospitals. Likewise, it is interesting to observe that younger groups with patterns associated with mental health and digestive problems (P6, P7, P8 or P11) had more visits to A&E units than other more complex patterns, particularly in low SES areas. Indeed, these were patterns characterised by developmental or digestive conditions in young people and children that may occasionally lead to increased use of these services[57,58]. These findings highlight the importance of a comprehensive and systematic analysis, such as that of our study, which in addition to characterising the different patterns of multimorbidity allows us to make this problem visible in groups that have not traditionally been studied.

Lastly, although the patterns with the strongest association with mortality were cardiovascular and complex patterns (i.e., combining cardiovascular, musculoskeletal, and mental health diseases), the transversality of the pattern of dependence and liver disease (P20) was also remarkable, as this is a pattern that is highly present in both the low and middle SES areas as well as lesser extent in the high SES areas. Therefore, in terms of mortality, this pattern is found to be relatively shared across different SES groups. Although this finding could be counterintuitive concerning the existing literature (particularly considering that addition problems are commonly associated with social and occupational class differences)[59,60], it should be noted that such a relationship may make sense if we consider that the main population centres of the province are linked to a robust wine industry that has traditionally employed an important sector of the population in these areas. Therefore, the cause of the transversality of mortality due to this pattern could be linked to the winemaking context of this area of southern Spain. However, in the absence of studies that corroborate our hypothesis, we will have to continue developing this line of research, observing how the area's specific characteristics can influence mortality associated with this multimorbidity profile.

Again, of particular interest was the distinctive P15 pattern (Hypertension + Dyslipidaemia + Arthrosis), which resurfaces as a noteworthy differentiating factor. Despite being one of the common disease profiles among older people, this highly prevalent pattern has a lower association with mortality compared to other patterns of even younger people, but only in mid/high SES areas. In fact, it is known that mortality associated with multimorbidity in low-resource areas is higher[6,41,61,62].

Our research boasts a major strength in its extensive dataset size, which encompasses data from the entire province of Cadiz. This comprehensive coverage provides a unique opportunity to characterise multimorbidity profiles even within smaller population units. Moreover, our analysis includes individuals across all age groups, leading to a significantly larger number of identified multimorbidity patterns compared to previous studies. However, it is important to acknowledge some limitations inherent to the study design.

First, since we relied on medical records, individual sociodemographic information was limited to sex and age. Therefore, the addition of more individual level data (e.g., socioeconomic variables such as education or occupational status) could have facilitated a better understanding of the differences associated with chronicity. Second, the lack of a clear consensus for the patterns naming in the literature may mean that the methodology used in our study is not fully comparable to that of other studies. However, considering the breadth of conditions in our work and the great difference in the prevalence of these pathologies according to age (such as hypertension in those over 65 years of age or dermatitis in those under 45), we believe that the segmentation strategy used through the LCA technique is the least likely to cause bias. Third, it is essential to note that, as an observational study, we cannot infer causality. Therefore, the findings emphasise the need to transition this research into a longitudinal study, allowing us to examine the temporal progression of patterns as individuals age. Having the date of diagnosis will allow us to characterise the chronicity profiles obtained through the analysis of multimorbidity trajectories. Fourth, it is also necessary to point out that cancer did not define any of the patterns that we obtained in the classification of the LCA, however, for future studies it would be necessary to go deeper into the relationship between the different types of cancer recorded and the resulting patterns. Finally, by including the years 2020 and 2021, we do not know to what extent COVID-19 may have specifically affected the association between multimorbidity, health service use and mortality. In this sense, although it is perhaps still early to give a conclusive answer along these lines it is necessary to address this problem through future work.

In conclusion, the 25 multimorbidity patterns identified demonstrate an important variability across this small area of Spain. Although the most prevalent and well-known patterns are adequately classified, the wide variety of chronic diseases analysed has allowed us to identify some combinations that are less visible in clinical practice (for example, the case of dependencies and their cross-cutting nature among social classes). By encompassing a population cohort spanning across minors to older adults, we have successfully characterised diverse multimorbidity profiles in both men and women throughout the life cycle. Moreover, our study goes beyond identifying disease pathways strongly linked to mortality, as we have also pinpointed geographical areas and service types exhibiting a heightened prevalence of specific patterns. While further research is necessary to understand the mechanisms behind the emergence of multimorbidity patterns, this study offers additional evidence in this area, which could help identify at-risk groups and inform locally tailored preventive measures based on local socioeconomic characteristics. Finally, these findings underscore the persistence of social inequalities in multimorbidity, which must be addressed to mitigate the impact of chronicity on patients' quality of life, healthcare utilisation, and mortality rates.

## Methods
### Design and setting
This study is observational, cross-sectional, and descriptive, relying on anonymized health records of all inhabitants of the province of Cadiz

without age restrictions. The province of Cadiz is a region in Southern Andalusia (Spain) with a population of around 1,250,000 inhabitants (approximately 15% of the province lives in rural areas)[63]. It has a universal health care system, but it is one of the provinces with the worst deprivation rates in Spain[64].

Specifically, the data for this study are sourced from the Population Health Database (PHD) of the Andalusian Health Service[65], a health information system that gathers clinical data and data regarding the utilisation of healthcare resources for all individuals residing in Andalusia (Spain). For this study, we have access to anonymized medical records pertaining to chronic conditions of all individuals residing in the province of Cadiz (Andalusia) until 2021, as well as their utilisation of healthcare services from 2014 to 2021. All this information was distributed across two files in which healthcare users were assigned (and subsequently linked) through a unique identifier. The first file contains information about pathologies that have been diagnosed at some point in the patient's medical history, while the other file contains information about each individual and their utilisation of healthcare resources during the specified period.

In our dataset, we initially had the medical records of 1,375,068 patients, which we transformed into 1,142,367 by discarding individuals with (1) zip code of origin not belonging to the province of Cadiz, (2) who had no health data in the last 3 years (2019–2021) and had not died, so they no longer lived in the province.

We obtained this data as a part of the DEMMOCAD (ITI-0028-2019) project, which aims to conduct an in-depth analysis of multimorbidity profiles in this geographical area. The Andalusian Biomedical Research Ethics Coordinating Committee (Comité Coordinador de Ética de la Investigación Biomédica de Andalucía) has authorised the DEMMOCAD project for using these medical records (PEIBA:2249-N-19).

## Measures and variables

The records available in the PHD have basic information on the individual sex, date of birth, zip code of origin and health centre of reference. With respect to chronic conditions, the Andalusian Health Service has included 80 pathologies for their interest, severity, duration and/or prevalence and a series of mechanisms have been articulated to make these pathologies easily identifiable. These pathologies are diseases that are of a chronic nature and are composed from ICD diagnoses (ICD9MC or ICD10ES)[65,66]. In this database, a patient is understood to have a given pathology at a given time if that patient has any of the diagnostic codes that are part of that pathology in effect at that time. These diagnostic codes come from various sources in the public health system (primary care, hospital consults, hospitalisations, emergency, and nursing offices). The process by which these pathologies are associated to diagnostic codes and the entire creation of the database has been validated by the Andalusian Health Service and transformed and adapted for use in research environments.

In our study, we used 63 of the 80 original pathologies from the mentioned database and an additional pathology, called "Other Cancer", which groups together those neoplasms that had a prevalence of less than 0.5% in the population. The decision to group these pathologies is related to the 1% prevalence cut-off in each stratum (detailed in the data analysis section), with the aim that some types of neoplasms may appear in the patterns that would otherwise be missed by the LCA technique. The 64 pathologies on which the patterns of this study are based and their prevalence in the population can be seen in the Supplementary Information (see Supplementary Report). An individual was considered to have one of the pathologies in the database if he or she appeared in any temporary record diagnosed with that chronic condition, so it can be considered that the baseline was taken as the beginning of each patient's medical history, but in this case, we only had information about the pathology and not the date of onset.

In addition, we have the use of public health services in the last 8 years and whether the patient has died. Health services use is divided into three categories. On the one hand, primary care visits or General Practitioner Visits (GPs), which in our health system are associated with routine visits for common symptoms, medication, or regular check-ups. These visits also include the usual paediatric visits and periodic childhood age controls (up to 14 years of age in Andalusia). Then hospital visits associated with consultations for medical specialities (such as cardiology, traumatology, or dermatology). Lastly, A&E visits, referring to hospital or health centre visits for health situations that are more urgent and probably more serious. The information on death is only cross-sectional, not being available in this study the date of death but the status of the patient until December 31, 2021.

The basic unit on which we will analyse the association of multimorbidity patterns with socioeconomic characteristics is the local health area of the individual through the zip code of residence. The province of Cadiz contains 113 unique zip codes, which were grouped into 68 to have at least 1000 individuals in each local area. Each zip code is associated with a municipality, with some municipalities having several zip codes, which divide the municipality into local health areas.

Considering the local health area as the unit of aggregation, the socioeconomic variables that we will use will have an ecological character, being associated with the entire area of interest. In the first place, we have the deprivation index (DI) by census area in Spain[67] is available to measure each local health area's social characteristics. This index is a summary of 14 socioeconomic characteristics of each census area. It is constructed so that census tracts with a DI < 0 indicate deprivation and social characteristics below the average of the areas in Spain. Since each census tract can be associated with a zip code, we can also determine the DI of each code within the province of Cadiz as the weighted average of all the DIs of the census tracts in that code. In addition, we have as second ecological variable the income data per individual and household for each zip code extracted from the National Institute of Statistics (https://www.ine.es), data that is not found within the 14 characteristics of the DI. These two ecological variables (income and DI) will be the base variables to define the SES of each local health area.

## Data analysis

The analysis was based on a large set of health records from the Andalusian Health Service, with a total initial sample of 1,142,367 individuals. Initially, we selected those individuals with multimorbidity (2 or more pathologies on the database). To simplify the analysis and considering the large differences in the prevalence, of chronic conditions due to sex and age, we stratified the database into groups according to sex and age (<16, 16–44, 45–64, 65–79 and >79), obtaining 10 groups.

Within each stratum, we streamlined the analysis by focusing on the 64 chronic conditions that exceeded a prevalence threshold of 1%. The decision to take this cut-off point had two main objectives: (1) to obtain models that consider only the chronic conditions that have a minimum prevalence in each sex and age group and, at the same time, (2) facilitate calculation time and convergence of LCA models. With this, we can obtain models with better goodness-of-fit indices and more parsimonious by making use of only those indicator variables that are relevant to the latent structure. By testing higher cut-off points (2%, 3% and 5%) we observed that the convergence of the models and goodness-of-fit indices did not improve particularly and several patterns that provided relevant information to our study were lost. After choosing these chronic conditions in each stratum, we exclusively considered individuals with multimorbidity based on these selected conditions. This process resulted in the formation of 10 sex and age groups with the following sample sizes: 8047; 11,789; 38,130; 50,817;

104,398; 88,420; 73,326; 60,129; 33,700; and 20,410. The cumulative number of individuals identified with multimorbidity across these groups amounted to 490,130.

We used LCA, a multivariate technique used to classify observations based on patterns of categorical responses, to identify multimorbidity patterns in the different sex and age subgroups (see Supplementary Information). LCA is a well-suited technique for our data, given the binary variables associated with chronic conditions and the causal relationship we hypothesised between the conditions in each pattern. It is a technique that is also one of the most widely used to detect multimorbidity patterns[15,16]. It is also capable of uniquely classifying everyone into a single multimorbidity pattern by analysing the probabilities of belonging to each chronicity profile, while choosing an optimal number of classes based on various diagnostic tests[68]. Unlike others techniques for defining multimorbidity patterns such as the O/E ratios or clustering techniques, the LCA technique made it possible to identify patterns by means of membership probabilities, while offering purely quantitative metrics (AIC, ABIC, CAIC) that allowed us to assess the validity of the selected patterns. In addition, as a commonly used method in recent studies on the classification of multimorbidity patterns, it is possible to obtain better comparisons with respect to recent evidence in this field[16].

Given the large sample size and the number of conditions, we computed the LCA models with the *POLCAParallel*[69,70] package, which implements the usual LCA using all processor cores, thus decreasing computation time. We established the number of appropriate patterns in each LCA model, considering three criteria[71]. Initially, we assessed the goodness-of-fit indices of the models, taking into consideration BIC, ABIC and CAIC. A lower value of these indices indicates a better fit of the model. Additionally, we examined the probability of membership in each class (i.e., multimorbidity pattern) and assessed the clinical interpretability of the results.

After the identification of the 25 multimorbidity classes, we performed a visual analysis of the differences in the patterns at the spatial level on maps of the province of Cadiz. To do this, we represented the relationship between the prevalence of the resulting patterns and the income per capita in each local health area in the province using a dual chromatic representation of these variables with the *biscale* package in R[72]. This allowed us to identify specific differences associated with certain areas with low income per capita.

Following this analysis, we grouped the local health areas according to their SES. Within the variables associated with each code, we took the deprivation index and the income per household as the status reference. We performed a cluster analysis on these two variables with the k-means technique. We obtained three different groups within the 68 local health areas, which we categorised as low SES (low DI and low income), medium SES (medium DI and medium income) and high SES (high DI and high income). The qualitative analysis of the groups obtained in the cluster analysis corresponds to our knowledge of the local health areas.

With these groupings, we analysed the differences in the prevalence of the 25 patterns in each age and sex stratum through the Kruskal–Wallis rank test. We also calculated the average use of health services (GP visits, hospital consults and emergencies) for each person over the 8 years and we tested the difference in the use of these services within each pattern and each stratum of the local health area through Kruskal–Wallis rank test. Finally, we tested the impact of multimorbidity patterns on mortality through a logistic regression analysis, taking mortality as the dependent variable and the multimorbidity pattern of each person, sex, and age as independent variables. We performed this regression analysis separately in each local health area associated with the SES groups.

We conducted all analyses and tables using RStudio and R (4.2.1). We took the standard 0.05 as the significance level and indicated those significance levels lower than 0.1.

## Reporting summary
Further information on research design is available in the Nature Portfolio Reporting Summary linked to this article.

## Data availability
In this research, the data have been obtained as part of the DEMMO-CAD project after the acceptance of the Andalusian Biomedical Research Ethics Coordinating Committee (Comité Coordinador de Ética de la Investigación Biomédica de Andalucía). The study data are available under restricted access for patient confidentiality and privacy concerns of the Population Health Database custodians (Andalusian Public Health System, SSPA), access can be obtained upon request to the Director of Health Care and Health Outcomes of the Andalusian Public Health System (SSPA) who can provide the data for the purpose of use in the health and research environment. The researcher must provide the necessary documents accrediting that the project and/or research to be carried out complies with data protection laws and the application will be submitted to a committee of the SSPA, which will decide whether or not to grant access to the data. The request for the use of the data must be addressed to the person responsible for the data (Director General de Asistencia Sanitaria y Resultados en Salud, Servicio Andaluz de Salud, Avd. de la Constitución no. 18, 41071, Seville, Spain), who will usually respond within 2 weeks and provide the data once it has passed through the ethics committee and has been favourably evaluated. Source data are provided with this paper.

## Code availability
The code used for this study is available on Zenodo (https://zenodo.org/records/10078002).

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

## Acknowledgements
We would like to acknowledge the support of the University Research Institute for Sustainable Social Development, the Biomedical Research and Innovation Institute of Cadiz (INiBICA), the University of Cadiz and the Ramon y Cajal programme run by the Spanish Ministry of Science and Innovation. This publication was supported by public funds by the ITI call (Integrated Territorial Investment), developed by the Health Department of the Andalusian Government (ITI-0028-2019). The DEMMOCAD project has been 80% co-financed by funds from the European Regional Development Fund (ERDF) operational programme of Andalusia 2014–2020. This publication and research has been partially granted by the INDESS (Instituto Universitario de Investigación para el Desarrollo Social Sostenible) and the University of Cadiz, Jerez de la Frontera, Spain.

## Author contributions
J.A.-G. led the DEMMOCAD project, obtained the data, conceived the study, interpreted the results, and developed the final version of the manuscript. J.C.-B. analysed the data and provided a first interpretation of the results, V.S.-L., E.O.-M. and B.R.-F. did the literature search and contributed to the final version of the document. The manuscript, figures, and final tables were read and approved by all authors.

## Competing interests
The authors declare no competing interests.
