## [Peer Review File · Nature Communications]

Epidemiology, mortality, and health service use of local-level multimorbidity patterns in South Spain.REVIEWER COMMENTS

Reviewer #1 (Remarks to the Author):

This is an original investigation looking at the distribution (by age, sex, geographical area, SES etc) of multimorbidity patterns in a Spanish population and its prognosis. There are a number of missing pieces of information that hamper the correct interpretation of the study's findings, plus there are methodological issues that are imprecise and debatable. Here my comments:

- In the manuscript there is an inappropriate use of the term incidence (eg, Figure 2). - Please, check carefully how this term was used.

- Showing local map distributions of a phenomenon is of interest for local policy makers and researchers. In a study like the present one – and in consideration of the target journal – I wonder what would be the added value of showing local maps. Local maps are interesting also in case ratios or at least two parameters (eg, multimorbidity and SES) are shown together, but this is not the case in this study. In addition, only a selection of the patterns are shown. My recommendation is to leave the maps as supplementary material and focus more on reporting measure of association (even if at the ecological level) between multimorbidity and other covariates. Current maps have incomplete legends and require more explanation.

- A table 1 with sample characteristics (especially considering the wide age range) is currently missing.

- In Figure 6 it looks like participants <16 years reported more than 10 GP visits per year? Isn't such a rate a bit high? Are individuals in the pediatric age included? Are they also seen by GPs? This should be made clearer for a correct interpretation of the findings.

- It is difficult to properly follow results when only a number (eg, P15) is attributed to the pattern.

- More information is needed regarding the health records part of this study. It is not clear from the description what they exactly cover. Are the authors using primary care records to derive diseases or primary care + other sources? This should be made clear.

- Regarding the disease assessment and categorization and multimorbidity patterns identification:

- 1) It is not clear how the authors ended up considering 64 (no more not less) conditions, are they referring to a specific and validated list of diseases?
- 2) Which diseases <1% were excluded from the LCA? Why a cutoff of 1% was chosen?
- 3) How did diagnoses were coded? ICD codes? Other? What is the reliability of such diagnoses? Any quality assessment performed prior to this study?
- 4) Looking at the disease list, it is not clear why, for example there is specific mention of given cancers (uterus, testicle) and then just an "other cancer" category. Disease aggregation and categorization represent a key step in patterns identification. First, depending on how diseases are fragmented or unified in categories people can be classified as multimorbid or not (especially in studies with very young people) and second, the clustering (also in LCA) depends a lot on this initial information. My suggestion is to carefully revise the list of diseases and make changes in order to fairly attribute similar weights to different classes, avoiding replications or rough aggregations. In this regard, there are available published lists of diseases.
- 5) It is not clear which time window prior to the baseline was interrogated to attribute diagnoses to the study participants. Any diagnoses prior the baseline piled up at baseline? Any diagnoses in a time frame prior to the baseline?
- 6) How the name of the patterns was chosen? I have the impression that disease prevalence was used, which is inaccurate. Patterns are usually named after the diseases over-expressed in each of them. Namely, diseases which observed prevalence in a given pattern is highly increased compared with the prevalence of the same disease in the rest of the sample. Observed/expected ratio and exclusivity are common used parameters to label the patterns.
- 7) The number of identified patterns is extremely higher than what found usually in similar papers. How is this put in context of other studies?

- Through logistic regressions the authors tested the association between the patterns and death. It is not clear on which time window cumulative mortality was considered. Was the baseline the same for all participants?

Reviewer #2 (Remarks to the Author):

Thank you for the opportunity to review this manuscript. This is a study of characterization of multimorbidity patterns in the region of Cadiz, Spain. The study was conducted using the region's universal healthcare system registry. Data are presented stratified by local health areas, age and gender. The study is very interested and well-conducted.

There are several strengths, such as the use of the health system registers in all ages, which shows interesting results, such as the presence of several groups of multimorbidity in younger ages, or the

observation of different patterns according to different SES local areas. I have some comments to help improve it.

Overall comments:

#1 In the objectives, it is mentioned that the multimorbidity patterns are characterized in a sample of 1,375,068 patients. Here, you might need to mention that these are people who have been in contact with the public health system during the period 2014-2021.

2 The language can be improved, and might benefit from a linguistic review

#3 It is unclear how you classified a person as having certain diagnose. If you are defining someone as having a certain chronic diagnose if the diagnose appears during the whole period (2014-2021), could it happen that one chronic condition appears several years before the other is diagnosed? If this is the case, then the multimorbidity classes you found might not adequately capture the risk that one diagnose has on the onset of another. For example, someone who was diagnosed of alcohol dependence in 2014 and then presented a new diagnose of liver disease in 2020. This needs to be clarified and discussed in the Discussion section.

Results

#4 Are the multimorbidity classes mutually exclusive (that is, one person is only classified in one of the groups)? Please specify.

#5 Additions appear several times through the manuscript. This is relevant. However, it is important to specify (at least in the annex) which diagnoses are included in each broader category. It would also be useful to analyze addictions as a separate entity, due to the significant expected contribution to chronic diseases in your region.

#6 It would be useful to include a table (as an annex) indicating the total number of cases per each multimorbidity pattern.

#7 Figure 5 shows the total number of visits, however it is difficult to know if the use of services is relatively high or not. For that, it would be useful to present percentages instead.

#8 If the authors have access to date of death, instead of using logistic regressions, it would be better to conduct Cox models to understand the risk of mortality. Otherwise, the authors need to be careful when talking about risk of mortality (instead, they are looking at associations, this should be changed throughout the manuscript).

Methods

#9 It is mentioned that Primary Care, Emergency units and specialized care registers were used. Did you also use data from hospitalizations? If so, please indicate it, or justify why you did not use it.

#10 Data from 2014-2021 were used for the present study. Apart from the diagnoses, did you also have access to date of diagnosis?

#11 How did you consider someone with a diagnosis? What happens with those participants who had a diagnosis prior the observational period (prevalent cases)? Were they considered in the analyses?

#12 Which diagnostic codes were used by the health system registers (CIE-9, CIE-10)? Are they unified during the whole observational period? Please specify. Additionally, please indicate (in the supplementary material as a Table), the CIE code that corresponds to each diagnosis used in your analyses.

#13 Are the registers digitalized? How they were linked? (using an identifier...) Please specify.

#14 How the income per person variable was obtained? Was it included in the registers?

#15 Where mortality data was obtained from? How was linked to your data?

Discussion

#16 I think it is worth to mention that the higher use of services in younger people with multimorbidity might reflect that they are not receiving a correct management for their conditions. This is an important finding of your study.

#17 COVID-19 has not been mentioned. However, this study includes data from 2020-2021. How do the authors think COVID-19 has affected your results, for example, in mortality? This needs to be discussed.

#18 Lines 310-312. I think this sentence needs to be re-written, since it is difficult to follow as it is.

#19 Paragraph from lines 334-346. This would only be possible to know if you can conduct a longitudinal study, please indicate this.

#20 Line 392. When the authors say "Thus, this pattern, associated with somewhat healthier ageing...", they need to be cautious, since we cannot state that P15 is related to healthier ageing. I would delete this sentence or re-formulate it.

#21 I miss more discussion on the mental pattern (P5). It is surprising that this class is associated with higher mortality, compared with cardiovascular. What are the reasons for this finding?

#22 The role of privately funded health care services need to be further discussed. Could use of privately funded health care services have led to some bias in your results, especially that concerning differences between high and low SES areas?

REVIEWER COMMENTS

Thank you very much for your comments. The following is a point-by-point response to the changes we have made in relation to each of the reviewers' comments and the respective proposals for improvement of our manuscript.

Reviewer #1 (Remarks to the Author):

This is an original investigation looking at the distribution (by age, sex, geographical area, SES etc) of multimorbidity patterns in a Spanish population and its prognosis. There are a number of missing pieces of information that hamper the correct interpretation of the study's findings, plus there are methodological issues that are imprecise and debatable. Here my comments:

- In the manuscript there is an inappropriate use of the term incidence (eg, Figure 2). - Please, check carefully how this term was used.

Many thanks for this observation. As recommended the term has been corrected.

- Showing local map distributions of a phenomenon is of interest for local policy makers and researchers. In a study like the present one – and in consideration of the target journal – I wonder what would be the added value of showing local maps. Local maps are interesting also in case ratios or at least two parameters (eg, multimorbidity and SES) are shown together, but this is not the case in this study. In addition, only a selection of the patterns are shown. My recommendation is to leave the maps as supplementary material and focus more on reporting measure of association (even if at the ecological level) between multimorbidity and other covariates. Current maps have incomplete legends and require more explanation.

Thanks for the comment. As the reviewer rightly points out the local maps may provide little information for more international readers. For this reason, in order to provide a new map of the spatial relationship between multimorbidity patterns and socioeconomic status, we have proceeded to show the combined relationship of the two variables (i.e., the specific multimorbidity pattern and average household income of the local area). In this way, we believe that the spatial relationship between the different classes of multimorbidity and income in the area is better depicted.

- A table 1 with sample characteristics (especially considering the wide age range) is currently missing.

As recommended we have incorporated a table 1 describing the main characteristics of the sample by sex group and (average) number of conditions according to the different age groups analyzed. Additionally, trying to provide a better characterization of the diseases, the top ten most prevalent conditions have been also included in this table.

- In Figure 6 it looks like participants <16 years reported more than 10 GP visits per year? Isn't such a rate a bit high? Are individuals in the pediatric age included? Are they also seen by GPs? This should be made clearer for a correct interpretation of the findings.

Thanks for this comment. As the reviewer has adequately observed this higher rate in among the participants <16 years may be, in part, due to the fact that routine pediatric GP visits are included in the data we have. In any case, observing the visits to A&E services we assume that we are also detecting differences that might be due to

incorrect management and care of these younger groups. Accordingly, we have included a sentence explaining this for a better interpretation of the results and subsequent findings.

- It is difficult to properly follow results when only a number (eg, P15) is attributed to the pattern.

Again, thank you very much for this observation. As rightly pointed out, the document can be difficult to follow when there are so many numerical references to multimorbidity patterns. In this sense, we have tried as far as possible to define each multimorbidity pattern in a nominal way. We hope that the new changes will contribute to a better reading and understanding.

- More information is needed regarding the health records part of this study. It is not clear from the description what they exactly cover. Are the authors using primary care records to derive diseases or primary care + other sources? This should be made clear.

As recommended, we have included additional information in the method section describing the dataset we have, the sources (i.e., health registries from the Andalusian Public Health System). In addition, for further information, links to the public health system platform are provided, where the data source is comprehensively described.

- Regarding the disease assessment and categorization and multimorbidity patterns identification:

1) It is not clear how the authors ended up considering 64 (no more not less) conditions, are they referring to a specific and validated list of diseases?

Thanks for this comment. We have expanded the method section by explaining that we used 63 of the original 80 pathologies from the aforementioned dataset and an additional pathology, called "Other Cancer", which grouped together neoplasms with a prevalence of less than 0.5% in the population. The decision to group these pathologies is related to the 1% prevalence cut-off in each stratum (detailed in the data analysis section), with the aim that some types of neoplasms may appear in the patterns that would otherwise be missed by the LCA technique. To provide more details of the conditions included, we have added a Supplementary Report with information on the 64 pathologies on which the patterns of this study are based and their prevalence in the population.

2) Which diseases <1% were excluded from the LCA? Why a cutoff of 1% was chosen?

Thanks for this observation. As we now explain in the method section, the decision to take the 1% cutoff point had two main objectives: (1) to obtain models that consider only the chronic conditions that are prevalent in each sex and age group and, at the same time, (2) facilitate calculation time and convergence of the models. With this cutoff point, we can obtain models with better goodness-of-fit indices and more parsimonious by making use of only those indicator variables that are clearly related to the latent structure. By testing higher cutoff points (2%, 3% and 5%) we observed that the convergence of the models and goodness-of-fit indices did not improve particularly and several patterns that provided relevant information to our study were lost.

3) How did diagnoses were coded? ICD codes? Other? What is the reliability of such diagnoses? Any quality assessment performed prior to this study?

Thanks for the comment. We have explained in the method section that “With respect to chronic conditions, the Andalusian Health Service has included 80 pathologies for their interest, severity, duration and/or prevalence and a series of mechanisms have been articulated to make these pathologies easily identifiable. These pathologies are diseases that are of a chronic nature and are composed from ICD diagnoses (ICD9MC or ICD10ES). A patient is understood to have a given pathology at a given time if that patient has any of the diagnostic codes that are part of that pathology in effect at that time. These diagnostic codes come from various sources in the public health system (primary care, hospital consults, hospitalizations, emergency, and nursing offices). The process by which these pathologies are associated to diagnostic codes and the entire creation of the database has been validated by the Andalusian Health System and transformed and adapted for use in research environments.”

4) Looking at the disease list, it is not clear why, for example there is specific mention of given cancers (uterus, testicle) and then just an “other cancer” category. Disease aggregation and categorization represent a key step in patterns identification. First, depending on how diseases are fragmented or unified in categories people can be classified as multimorbid or not (especially in studies with very young people) and second, the clustering (also in LCA) depends a lot on this initial information. My suggestion is to carefully revise the list of diseases and make changes in order to fairly attribute similar weights to different classes, avoiding replications or rough aggregations. In this regard, there are available published lists of diseases.

The choice to group other types of cancer –independently of the organ group– was motivated by the need to have a certain level of prevalence in order for the LCA technique to provide better pattern classifications. While initially a more logical organization in terms of cancer type would be sought, it was not possible to obtain a minimum prevalence (1% cutoff) that would result in cancer-specific patterns or patterns in which cancer would play a significant role.

Although due to the lower prevalence of this disease group we could not find a cancer specific multimorbidity pattern, as the reviewer rightly points out, this is an issue that needs further attention. Therefore, we have incorporated this observation in the limitations paragraph of the discussion section. Many thanks.

5) It is not clear which time window prior to the baseline was interrogated to attribute diagnoses to the study participants. Any diagnoses prior the baseline piled up at baseline? Any diagnoses in a time frame prior to the baseline?

We have explained this question in the methods section. Specifically, we have explained that an individual was considered to have one of the pathologies in the database if he or she appeared in any temporary record diagnosed with that chronic condition, so it can be considered that the baseline was taken as the beginning of each patient’s medical history. Many thanks for the comment.

6) How the name of the patterns was chosen? I have the impression that disease prevalence was used, which is inaccurate. Patterns are usually named after the

diseases over-expressed in each of them. Namely, diseases which observed prevalence in a given pattern is highly increased compared with the prevalence of the same disease in the rest of the sample. Observed/expected ratio and exclusivity are common used parameters to label the patterns.

Undoubtedly one of the most contentious points in research on multimorbidity patterns is how to label these patterns. In fact, there is currently no consensus in the literature on the best techniques for selecting multimorbidity patterns. In health research, multimorbidity has been “derived by different statistical methodologies, such as observed to expected ratios or odds ratios among the most commonly dyads or triads of chronic conditions, or cluster and factor analyses to identify systematic clusters among diseases” (Rajoo et al, 2021).

Although this was a topic of debate in our team at the beginning of the study in the face of diverse evidence, the experience of the working group in classifying multimorbidity patterns, and previous evidence from the literature in this field, we decided that the LCA technique was the most appropriate for a dataset with a wide diversity of pathologies (some with low prevalence), being today one of the most used techniques (Alvarez-Galvez et al. 2023). This technique, unlike others such as O/E or clustering techniques, made it possible to identify patterns by means of membership probabilities, while offering purely quantitative metrics (AIC, ABIC, CAIC) that allowed us to assess the validity of the selected patterns. Therefore, this approach is better than conventional clustering and O/E ratios methods because LCA employs probability-based classification methods to choose an optimal number of classes based on various diagnostic tests.

- Rajoo, S. S., Wee, Z. J., Lee, P. S. S., Wong, F. Y., & Lee, E. S. (2021). A systematic review of the patterns of associative multimorbidity in Asia. *BioMed Research International*, 2021.
- Álvarez-Gálvez, J., Ortega-Martín, E., Carretero-Bravo, J., Pérez-Muñoz, C., Suárez-Lledó, V., & Ramos-Fiol, B. (2023). Social determinants of multimorbidity patterns: A systematic review. *Frontiers in Public Health*, 11, 1081518.
- Park, B., Lee, H. A., & Park, H. (2019). Use of latent class analysis to identify multimorbidity patterns and associated factors in Korean adults aged 50 years and older. *PloS one*, 14(11), e0216259.

7) The number of identified patterns is extremely higher than what found usually in similar papers. How is this put in context of other studies?

Unlike other studies, we include groups of all ages and separate the analysis into age and sex groups, which means that certain patterns that would not appear in a global analysis do appear in the group analysis. The rationale for this is associated with the LCA technique, which recommends performing a differential group analysis if the measures of indicators (in our case, chronic conditions) are so high between different groups that the latent variables may be structurally different.

- Through logistic regressions the authors tested the association between the patterns and death. It is not clear on which time window cumulative mortality was considered. Was the baseline the same for all participants?

Thank you for this observation. The information on death is only cross-sectional, not being available in this study the date of death but the status of the individual until December 31, 2021. We have added this information in the document.

Reviewer #2 (Remarks to the Author):

Thank you for the opportunity to review this manuscript. This is a study of characterization of multimorbidity patterns in the region of Cadiz, Spain. The study was conducted using the region's universal healthcare system registry. Data are presented stratified by local health areas, age and gender. The study is very interested and well-conducted.

There are several strengths, such as the use of the health system registers in all ages, which shows interesting results, such as the presence of several groups of multimorbidity in younger ages, or the observation of different patterns according to different SES local areas. I have some comments to help improve it.

Overall comments:

#1 In the objectives, it is mentioned that the multimorbidity patterns are characterized in a sample of 1,375,068 patients. Here, you might need to mention that these are people who have been in contact with the public health system during the period 2014-2021.

Thanks for this observation. We have corrected this using the term "dataset" instead of "sample" and modified some sentences to clarify that we are working with all the health records from the public health system until 2021 and their use of health services from 2014 to 2021.

2 The language can be improved, and might benefit from a linguistic review

The language has been reviewed. Many thanks.

#3 It is unclear how you classified a person as having certain diagnose. If you are defining someone as having a certain chronic diagnose if the diagnose appears during the whole period (2014-2021), could it happen that one chronic condition appears several years before the other is diagnosed? If this is the case, then the multimorbidity classes you found might not adequately capture the risk that one diagnose has on the onset of another. For example, someone who was diagnosed of alcohol dependence in 2014 and then presented a new diagnose of liver disease in 2020. This needs to be clarified and discussed in the Discussion section.

Thank you for this comment. In relation to this question we have clarified in the document that in this research we only work with the diagnosis that the person had of a certain chronic condition and not with the date of appearance of this, so our research is of a cross-sectional nature and it was not our objective to determine the form of appearance of the pattern but to characterize which patterns of multimorbidity were present in the population as of December 2021. We have clarified in the methods section that the information in the period 2014 to 2021 refers to the use of services, which in the same way is only related cross-sectionally with the resulting multimorbidity patterns.

Results

#4 Are the multimorbidity classes mutually exclusive (that is, one person is only classified in one of the groups)? Please specify.

Yes, they are exclusive. The LCA technique allows us to classify according to the probability of belonging to which exclusive class each individual is assigned. We have clarified this aspect in the methodology of the article. Thank you very much for the observation.

#5 Additions appear several times through the manuscript. This is relevant. However, it is important to specify (at least in the annex) which diagnoses are included in each broader category. It would also be useful to analyze addictions as a separate entity, due to the significant expected contribution to chronic diseases in your region.

While we understand the reviewer's observation, in our data we do not have the diagnostic codes that define the diseases but only the pathologies that have been previously identified and subsequently recorded by the public health system. We have described this aspect in the method section. In any case, as we indicate in our paper, this is a line that, as the reviewer rightly points out, should be addressed in the future in order to understand the relationship between addictions and certain patterns of multimorbidity.

#6 It would be useful to include a table (as an annex) indicating the total number of cases per each multimorbidity pattern.

As recommended, we have included a table (as an annex) indicating the total number of cases per each multimorbidity pattern.

#7 Figure 5 shows the total number of visits, however it is difficult to know if the use of services is relatively high or not. For that, it would be useful to present percentages instead.

Figure 5 shows the average number of visits to health services for patients with multimorbidity. In any case, to avoid possible misunderstandings, we have modified the title of the figure and the text of the previous paragraph to make it clear that we are describing average values by sex, age and SES area.

#8 If the authors have access to date of death, instead of using logistic regressions, it would be better to conduct Cox models to understand the risk of mortality. Otherwise, the authors need to be careful when talking about risk of mortality (instead, they are looking at associations, this should be changed throughout the manuscript).

As we did not have the date of death data, we were unable to carry out a Cox regression, although we understand that this is work that we may be able to address in the future. To avoid problems of interpretation, we have eliminated the references we made when talking about mortality risks and introduced new references to possible associations.

Methods

#9 It is mentioned that Primary Care, Emergency units and specialized care registers

were used. Did you also use data from hospitalizations? If so, please indicate it, or justify why you did not use it.

What we have is visits to primary care, visits to hospital consultations (specialized care) and access to the emergency department, but we do not have data on hospitalizations. We have specified this aspect in the paper.

#10 Data from 2014-2021 were used for the present study. Apart from the diagnoses, did you also have access to date of diagnosis?

The date of diagnosis was not available to us as the initial cross-sectional design did not request this information from the health service. Thus, we mention this limitation in the discussion section of the article and refer to the need to start working with diagnosis dates to analyze multimorbidity trajectories.

#11 How did you consider someone with a diagnosis? What happens with those participants who had a diagnosis prior the observational period (prevalent cases)? Were they considered in the analyses?

Thank you for the comment. What we have considered are the chronic conditions that each person has been diagnosed with up to the year 2021, the data from 2014 to 2021 are exclusively from the service use dataset that we have linked to perform the analysis. We have clarified this aspect in the document. Moreover, the diagnoses are not defined by us but have been classified and validated by the team of public health service statisticians who have designed the Population Health Database. We have added more information on this resource in the methods section of our article.

#12 Which diagnostic codes were used by the health system registers (CIE-9, CIE-10)? Are they unified during the whole observational period? Please specify. Additionally, please indicate (in the supplementary material as a Table), the CIE code that corresponds to each diagnosis used in your analyses.

Thank you for this observation. We have explained in the methods section that the public health system has managed to classify the different pathologies by using the ICD9 and ICD10 (depending on the existing information in the different health registries). This information is explained in a report of the Health Population Base of the Andalusian health system, so we have included a link to this information on the website of this research resource.

#13 Are the registers digitalized? How they were linked? (using an identifier...) Please specify.

In response to the reviewer's doubts, we have improved the description of the procedure followed to link the different data sets (chronic pathologies, health services and mortality) from a unique identifier that was anonymized by the IT staff of the public health services themselves.

#14 How the income per person variable was obtained? Was it included in the registers?

Thanks for this comment. Individual income information was not available, so we opted to use the average income per capita data by small health area. In this way, we have made it clear in the article that it is a data that allows us to describe a relationship at the ecological level, but not at the individual level.

#15 Where mortality data was obtained from? How was linked to your data?

As previously mentioned, this has been described in the methods section. Many thanks.

Discussion

#16 I think it is worth to mention that the higher use of services in younger people with multimorbidity might reflect that they are not receiving a correct management for their conditions. This is an important finding of your study.

As recommended by the reviewer, we have included this observation emphasizing the need to tackle these details in specific studies addressing the problem of multimorbidity in groups that may be encountering difficulties in managing multi-pathology through the medical pathway.

#17 COVID-19 has not been mentioned. However, this study includes data from 2020-2021. How do the authors think COVID-19 has affected your results, for example, in mortality? This needs to be discussed.

Many thanks. As recommended, this limitation has been mentioned in the discussion section.

#18 Lines 310-312. I think this sentence needs to be re-written, since it is difficult to follow as it is.

The language has been reviewed and complex sentences have been rewritten. Many thanks.

#19 Paragraph from lines 334-346. This would only be possible to know if you can conduct a longitudinal study, please indicate this.

We have corrected any comments that could be confusing in relation to the study design used and the possible conclusions that could be drawn from it. We have tried to better specify that we are dealing with a cross-sectional study in which we can only describe and measure associations.

#20 Line 392. When the authors say "Thus, this pattern, associated with somewhat healthier ageing...", they need to be cautious, since we cannot state that P15 is related to healthier ageing. I would delete this sentence or re-formulate it.

As recommended these sentences has been deleted. Many thanks.

#21 I miss more discussion on the mental pattern (P5). It is surprising that this class is associated with higher mortality, compared with cardiovascular. What are the reasons for this finding?

Thank you for this observation, in relation to this issue we must consider that if we take all cardiovascular conditions as a whole they would have more weight than mental health conditions, however, cardiovascular problems present a wide combinatory with other pathologies and their explanatory power is reduced when disaggregated into different groups. We have mentioned this aspect in the discussion.

#22 The role of privately funded health care services need to be further discussed. Could use of privately funded health care services have led to some bias in your results, especially that concerning differences between high and low SES areas?

Thank you for this comment. Unlike other regions in Spain, private health services in Andalusia are in the minority, so our data are covering most of the population. Moreover, in Andalusia we have a single clinical record per patient that incorporates data from both private and public services. Of course, there may be routine visits to the private doctor that are not recorded, but the usual practice is that when there is a diagnosis it is recorded in the patient's single clinical history for subsequent follow-up through the public health system. We have extended and improved this aspect in the discussion section.

REVIEWERS' COMMENTS

Reviewer #1 (Remarks to the Author):

I have no further comments

Reviewer #2 (Remarks to the Author):

The comments and reviews have been satisfactorily addressed by the authors.

Thank you

REVIEWER COMMENTS

Thank you very much for all the comments and suggestions for improvement, from our team we believe that all the changes made so far have contributed to enhance the final quality of our work.

Reviewer #1 (Remarks to the Author):

I have no further comments

Many thanks for all the comments and recommendations.

Reviewer #2 (Remarks to the Author):

The comments and reviews have been satisfactorily addressed by the authors.

Many thanks for all the comments and recommendations.